# Direct Motion Models for Assessing Generated Videos

**Kelsey Allen** [*1]   **Carl Doersch** [1]   **Guangyao Zhou** [1]   **Mohammed Suhail** [2]   **Danny Driess** [1]   **Ignacio Rocco** [1]
**Yulia Rubanova** [1]   **Thomas Kipf** [1]   **Mehdi Sajjadi** [1]   **Kevin Murphy** [1]   **Joao Carreira** [1]   **Sjoerd van Steenkiste** [*2]

## Abstract

A current limitation of video generative video models is that they generate plausible looking frames, but poor motion — an issue that is not well captured by FVD and other popular methods for evaluating generated videos. Here we go beyond FVD by developing a metric which better measures plausible object interactions and motion. Our novel approach is based on auto-encoding point tracks and yields motion features that can be used to not only compare distributions of videos (as few as one generated and one ground truth, or as many as two datasets), but also for evaluating motion of single videos. We show that using point tracks instead of pixel reconstruction or action recognition features results in a metric which is markedly more sensitive to temporal distortions in synthetic data, and can predict human evaluations of temporal consistency and realism in generated videos obtained from open-source models better than a wide range of alternatives. We also show that by using a point track representation, we can spatiotemporally localize generative video inconsistencies, providing extra interpretability of generated video errors relative to prior work. An overview of the results and link to the code can be found on the project page: `trajan-paper.github.io`.

---

[*]Equal contribution   [1]Google DeepMind [2]Google Research. Correspondence to: Kelsey Allen <krallen@google.com>, Sjoerd van Steenkiste <svansteenkiste@google.com>.

Author contributions: KA conceptualized the idea of using track-based latent motion features to measure video quality, moving from distributional metrics to per-video or video-video metrics. KA, SVS co-led the project and ran all the experiments. GZ invented an initial prototype of the TRAJAN architecture, which was further developed by CD. MS supplied the WALT checkpoints. All authors advised on the project direction and contributed to writing the paper.

*Proceedings of the $42^{nd}$ International Conference on Machine Learning*, Vancouver, Canada. PMLR 267, 2025. Copyright 2025 by the author(s).

## 1. Introduction

As generative video models become increasingly capable, the community needs more powerful methods for automatically evaluating the quality of generated videos. State-of-the-art models are getting better at generating what appear to be plausible looking frames, yet they still struggle to put together coherent motion (Brooks et al., 2024). The current gold standard for assessing video quality is collecting human judgments, but these are expensive to obtain and not scalable as a metric for regularly measuring improvements in modeling capabilities, e.g. throughout training.

Existing metrics, such as those based on Fréchet Video Distance (FVD) (Unterthiner et al., 2018; Ge et al., 2024; Luo et al., 2025), can capture certain elements of plausibility, but are more sensitive to frame-level content effects (Ge et al., 2024), and depend on access to the underlying training distribution which is not always available. Further, these approaches cover only one way of evaluating generated videos, i.e. by comparing entire distributions, and it remains unclear how to evaluate pairs of videos, or individual videos, with this approach.

Here we propose a new method that we show addresses many of these issues, by directly modelling 2D video motion. Given a single generated video, we estimate low-level, temporally-extended motion features as point tracks using the publicly available BootsTAPIR model (Doersch et al., 2024). Next, we (auto)encode these features to obtain dense high-level motion features using a novel *TRAJectory AutoeNcoder (TRAJAN)* architecture. We can then use the TRAJAN latent space to compare distributions of videos (as few as one generated and one real, or as many as two datasets), or the reconstruction error from TRAJAN to estimate per-video motion inconsistencies.

Point tracking, by design, separates the semantics from the motion content by focusing on features necessary for predicting motion without reconstructing the whole scene (Figure 1). This means that it will likely focus on plausible motion irrespective of semantic information. We show that this is indeed the case across the three major ways in which the community generally uses metrics to evaluate videos:

1. **For individual videos**, we measure how well TRA-

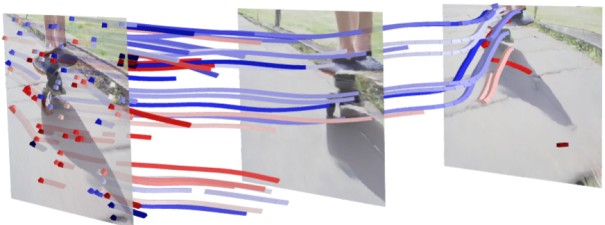

(a) Unrealistic object appearance, but plausible motion.

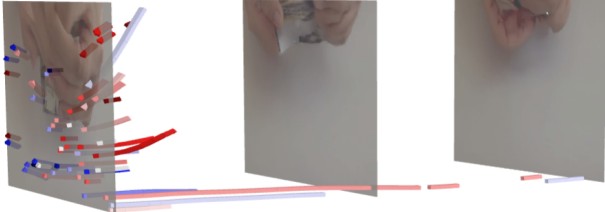

(b) Realistic looking hands, but which change appearance from frame-to-frame, with fingers morphing in and out of existence.

*Figure 1.* Predicted point trajectories from TRAJAN, colored by their reconstruction error with respect to BootsTAPIR (well reconstructed track, poorly reconstructed track) (Doersch et al., 2024). See `trajan-paper.github.io` for videos. The video in (a) has plausible motion but implausible frame-level appearance, while the video in (b) has plausible frame-level appearance but implausible motion (the fingers morph and disappear between frames). TRAJAN correctly predicts this discrepancy by focusing on *motion* irrespective of *appearance*.

JAN is able to reconstruct the original input tracks (§5.1). We find that this reconstruction score is more or equally predictive of human ratings for realism, appearance, and motion consistency of generated videos from 10 different open-source models than a wide range of alternatives (both appearance- and motion-based).

2. **For pairs of videos**, we compute a distance between their TRAJAN embeddings (§5.2), which better captures similarities in motion even when appearance-based pixel-level metrics suggest that they are different.

3. **At the distribution level**, we show that among several other choices — including VideoMAE v2 (Wang et al., 2023b), I3D (Carreira & Zisserman, 2017), and motion histograms (Liu et al., 2024a) — the features learned by TRAJAN are markedly more sensitive for detecting synthethic temporal distortions (§5.3).

As an additional contribution, we conduct a detailed human study on generated videos. We collect a new dataset of ratings that focuses on four aspects of motion: quality of interactions, the realism of the objects and behavior, consistency of appearance and motion, and the speed of objects and the camera. While acknowledging these are somewhat subjective, we find that TRAJAN typically outperforms prior methods at predicting human ratings, although with room for improvement, suggesting an area for further research.

## 2. Related Work

Metrics for assessing various aspects of video generation quality can broadly be categorized into a) operating on individual videos without a reference, b) comparing a generated video to a real "ground truth" reference, and c) comparing a distribution of generated videos with a reference distribution. Though these areas are generally considered separately, here we propose TRAJAN as a unifying model for measuring motion consistency in all of these three settings.

**Metrics for individual videos.** Several recent metrics have been proposed for evaluating text-to-video models on a per-video basis, since in this case a reference video is typically unavailable. However, a major focus has been on the semantic alignment between the prompt and its generated video. CLIPScore (Hessel et al., 2021) utilizes CLIP (Radford et al., 2021) to compare individual video frames with the text in the embedding space (Gu et al., 2024; Liu et al., 2024b; Wu et al., 2023). The main issue with CLIPScore is that it mainly captures semantics of individual frames, instead of dynamic properties of videos such as consistency over time, physical plausibility, or motion quality (Bansal et al., 2024b). Even for image generation tasks, it has been shown that such metrics are not always aligned with human judgement (Otani et al., 2023). Beyond CLIP, other approaches propose to use vision-language models (VLMs) to judge the semantic adherence between the prompt and the video (Wu et al., 2024; Bansal et al., 2024b), either zero-shot or by finetuning a VLM on human annotations of video quality (Huang et al., 2024). DOVER (Wu et al., 2022b) is a model trained to predict the average human subjective perception quality of a video. In addition to semantic adherence, Video-Phy by Bansal et al. (2024b) evaluate videos by whether they follow physical commonsense by finetuning a VLM (Bansal et al., 2024a) on this task.

Compared to these metrics, our approach is *not* focused on semantic adherence between a prompt and a video, but rather on the motion and appearance consistency of a video *independent of the prompt*. Our approach based on point tracks inherently takes the temporal evolution of the frames into account, which we show is crucial for matching human judgements of video quality on a per-video basis.

**Metrics for paired videos.** Evaluating the quality of generated video against ground truth frames often involves pixel-wise and perceptual metrics, such as Peak Signal-to-Noise Ratio (PSNR), Structural Similarity Index (SSIM), and Learned Perceptual Image Patch Similarity (LPIPS). PSNR measures pixel-level discrepancies using Mean Squared Error, but its pixel dependency makes it overly sensitive to

small, often imperceptible variations. SSIM (Wang et al., 2004) improves on this by comparing local patterns in brightness, contrast, and structure to better align with human perception. LPIPS (Zhang et al., 2018) uses a pre-trained deep networks (*e.g.*, VGG (Simonyan & Zisserman, 2014), AlexNet (Krizhevsky et al., 2012)) to compute similarity based on feature maps, capturing higher-level perceptual cues. While these metrics provide valuable insights into frame-level similarity, they evaluate each frame independently and thus fail to capture temporal coherence—an essential component of video realism.

**Metrics for video distributions.** FVD (Unterthiner et al., 2018) is a standard metric to compare a distribution of real world videos with generated ones (Bugliarello et al., 2024). However, Ge et al. (2024) show that FVD is biased towards the content of individual frames, possibly because of how the underlying I3D (Carreira & Zisserman, 2017) feature extractor was trained. Ge et al. (2024) propose to use VideoMAE features to address this, while concurrent work by Luo et al. (2025) considers a number of alternative appearance-based feature representations. Liu et al. (2024a) also propose a new feature representation, but they compute histogram-based motion features derived from estimated point tracks. Our approach is closely related in that we also use point tracks, however, a key difference is that we use learned TRAJAN features, which we find to perform markedly better. Additionally, we demonstrate how TRAJAN can be used to assess the quality of individual videos.

**Benchmarks & generative video models.** The field of generative video models is rapidly evolving. Most current methods for text-to-video models are either based on diffusion (He et al., 2022; Ho et al., 2022b;a; Singer et al., 2022; Luo et al., 2023; Wang et al., 2023a; Zhou et al., 2022; Khachatryan et al., 2023; Blattmann et al., 2023b) or transformers (Villegas et al., 2022; Wu et al., 2021; Hong et al., 2022; Wu et al., 2022a). Hence, there have been several efforts to evaluate such generative models (Liu et al., 2024d; Huang et al., 2024). VideoPhy (Bansal et al., 2024b) is a benchmark that focuses on prompts where a model has to obey the laws of physics, such as marbles rolling down a slanted surface. We evaluate TRAJAN both on VideoPhy (Bansal et al., 2024b) and EvalCrafter (Liu et al., 2024c).

## 3. Methods

To obtain metrics for evaluating motion in videos, we focus on two aspects: how to extract latent representations from videos, and how to compute ordinal-valued metrics for videos that can directly be understood as ranking individual videos as being better or worse. We propose TRAJAN, which gives us latent representations and can be used directly as a metric on a per-video basis.

For distribution-level comparisons, we compute the Fréchet distance between the latents in two datasets $P_R$ and $P_G$ as in prior work (Unterthiner et al., 2018). It is defined as $d(P_R, P_G) = min_{X,Y} E|X - Y|^2$ which simplifies to $d(P_R, P_G) = |\mu_R - \mu_G|^2 + Tr(\sum_R + \sum_G - 2(\Sigma_R \Sigma_G)^{\frac{1}{2}})$ when $P_R$ and $P_G$ are multivariate Gaussians. For video pair comparisons, we simply take the $L_2$ distance between the latents. For single video evaluations, we use the ordinal number (such as reconstruction error) to determine quality.

### 3.1. TRAJAN

Point tracking models like PIPs++ (Zheng et al., 2023) or BootsTAPIR (Doersch et al., 2024) can track points in arbitrary videos surprisingly well. Point tracks are inherently linked to *motion* rather than *appearance*, making them a strong candidate for assessing motion quality in videos. However, a set of point tracks is inconvenient to work with because the tracks are orderless, making it difficult to compare two sets without a ground-truth reference. Further, we must also contend with missing data due to occlusions.

To address this problem, we introduce the TRAJectory AutoeNcoder (TRAJAN, Figure 2). At a high level, TRAJAN is trained to reconstruct point tracks starting from random queries across space and time, and can provide both a latent representation and a reconstruction score. TRAJAN operates on a set of point trajectories $S = \{s_{t,j}\}$ coming from, e.g., BootsTAPIR, where $s_{t,j} = (x_{t,j}, y_{t,j}, o_{t,j})$ corresponds to $x$ and $y$ positions and occlusion flag $o$ at time $t$ for the $j$th trajectory, and is trained to reconstruct a separate set of query trajectories $Q = \{q_{t,j}\}$ similarly randomly sampled from the video.

**Architecture.** For a given track $j$, we first embed all locations $(x_{t,j}, y_{t,j}, t)$ with a sinusoidal embedding across space and time and project to $C$ channels. Then we add a "readout" token of length $C$, and perform self-attention to all tokens for track $j$, using $(1-o_{t,j})$ as an attention mask, which helps the representation be invariant to occluded points. After self attention, we discard all tokens except the "readout" token to obtain a fixed-length $C$-channel representation of each track. Next, we use a Perceiver (Jaegle et al., 2021)-style approach to encode all track tokens, namely by cross-attending from a learned set of 128 latent tokens to all track readout tokens, followed by self-attention. Finally, we project the latent tokens to a lower dimension resulting in a fixed-size $128 \times 64$ dimensional representation $\phi_S$ of the tracks.

To ensure this representation encodes the dense motion, we train it to reconstruct tracks from the video using a decoder. However, since we want to represent the motion as a dense field, we want it to be *invariant* to the specific query points used to obtain the support tracks. To achieve this, we let the decoder reconstruct held-out tracks not included in the

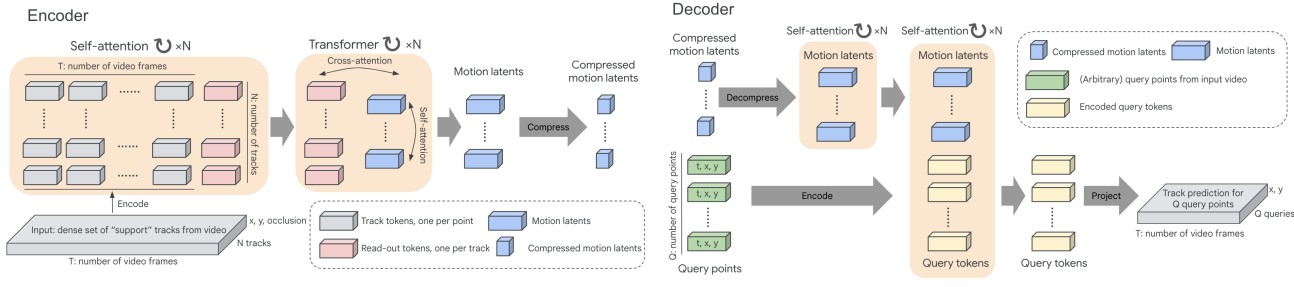

(a) Point trajectory encoder

(b) Query point decoder

*Figure 2.* **TRAJAN** – The trajectory encoder encodes a (variable-sized) set of point trajectories $(x_{t,j}, y_{t,j}, o_{t,j}, t)$ into a compressed motion latent $\phi_S$ of fixed size using a Perceiver (Jaegle et al., 2021)-style transformer architecture. The occlusion flag $o$ is used in the attention mask, making the representation invariant to occluded points. The decoder takes this latent $\phi_S$ and predicts for a query point $(x_q, y_q, t_q)$ the point track that goes through this point at all other times, as well as their occlusion flag. By training the autoencoder on different input and query points, the model learns to represent a dense motion field.

input. That is, the decoder takes $\phi_S$ and a query point $(x_q, y_q, t_q)$, and outputs the track that goes through this point. We up-project the tokens in $\phi_S$ to a higher dimension with an operator $U$, and compute a "readout" token via a sinusoidal position encoding of $(x_q, y_q, t_q)$. Next, we apply self-attention to the full token set, and discard all but the readout token. We finally apply a linear projection to the readout token to obtain $(x_t^q, y_t^q, o_t^q)$ for every frame, where $o_t^q$ is an occlusion logit trained by sigmoid cross-entropy, and $x_t^q, y_t^q$ are trained with an $L_1$ loss. In Appendix B.2.1, we provide more details about training and architecture.

**Average Jaccard.** An advantage of TRAJAN is that it not only computes representations that describe the motion contained in a video, but that it can also be used to evaluate the motion directly: the model was trained on real motions, so we expect unrealistic motions to be more difficult to reconstruct. Thus, we propose Average Jaccard (Doersch et al., 2022) (AJ) to measure the accuracy of the autoencoder reconstructions relative to the tracks it receives as input. Average Jaccard combines occlusion and position accuracy. For a given threshold $\delta$ Jaccard$_\delta$ considers "true positives" $(TP)$ to be predictions which are within $\delta$ of the ground truth. "False positives" $(FP)$ are predictions that are farther than $\delta$ from the ground truth (or the ground truth is occluded), and "false negatives" $(FN)$ are ground truth points where the prediction is farther than $\delta$ (or occluded). Jaccard$_\delta$ is $TP/(TP + FP + FN)$, and Average Jaccard averages Jaccard$_\delta$ over several thresholds (see also Appendix B.2.1).

### 3.2. Alternative models

#### 3.2.1. MOTION-BASED

For comparison to TRAJAN, we consider two alternative motion-based metrics: one based on *histograms* over point track trajectories, and one based on optical flow.

**Motion histograms** Liu et al. (2024a) propose to evaluate motion consistency in generated videos by estimating velocity and acceleration from point tracks. Inspired by HOG features (Dalal & Triggs, 2005), they partition the resulting volumes into $4 \times 5 \times 5$-sized tubelets and accumulate the magnitude of the values at each angle (using 8 bins) within a tubelet. Motion features are obtained by concatenating the resulting 1D histograms obtained for each tubelet using both velocity and acceleration. We apply this approach to 16-frame chunks of $64 \times 64$ densely sampled point tracks obtained from BootsTAPIR (Doersch et al., 2024) to yield a 9216-dimensional vector describing the motion within the corresponding 16-frame video.

**Optical flow** Another natural candidate for evaluating motion in videos is to make use of Optical Flow. Unlike point tracks, flow is usually estimated between consecutive frames, which might affect its ability to consider long-term motion patterns. Prior work has examined the *warping error* as a loss to make videos more temporally consistent (Lai et al., 2018), and was used more recently as a metric of temporal consistency in generated videos (Liu et al., 2024c).

For a given pair of frames, the warping error is obtained by computing the pixel-wise difference between the second frame and the "warped prediction" computed using the optical flow prediction applied to the first frame, while also accounting for predicted points of occlusion. Following Liu et al. (2024c), we calculate the warp differences on every two frames, and average across pairs to obtain the final score. To estimate optical flow we use a version of RAFT (Teed & Deng, 2020) with added improvements (Sun et al., 2022; Saxena et al., 2024) (see Appendix B for details). RAFT iteratively updates a flow field based on multi-scale 4D correlation volumes computed from learned features for all pairs of pixels. Due to the size of these volumes, we only consider the negative warping error (such that higher error corre-

sponds to a lower ordinal value) as a per-video metric as calculating a covariance matrix would otherwise be intractable.

### 3.2.2. APPEARANCE-BASED

Inspired by FVD, we also consider a number of methods that operate directly on RGB. Although such methods are freely able to focus on content or motion related information, prior work encountered a content-bias (Ge et al., 2024), especially for classification models.

**I3D** The I3D network is an inflated 3D convolutional neural network based on the Inception architecture for image classification that can be applied to videos (Carreira & Zisserman, 2017). Unterthiner et al. (2018) proposed to use the logits of an I3D network that was trained to perform action recognition on the Kinetics-400 dataset (YouTube videos of humans performing various actions (Carreira & Zisserman, 2017)) as a feature space for comparing videos. For a single-video metric from I3D, we use the negative entropy over the action class predictions (meaning lower scores indicate worse quality).

**VideoMAE** VideoMAE (Tong et al., 2022) is a Masked Autoencoder (MAE) trained on videos using a self-supervised reconstruction objective, which was previously found to reduce frame-level bias relative to I3D when used for evaluating videos (Ge et al., 2024). Here we use VideoMAE-v2 (Wang et al., 2023b), which is trained on a mixed set of raw video datasets. Importantly, the reconstruction objective for VideoMAE-v2 is not autoregressive – it is not trained to predict the future, rather it is trained to "fill in" missing patch tubes from a video. In practice, VideoMAE-v2 is usually further fine-tuned on specific downstream tasks to deal with domain shift.

We consider two variants of VideoMAE-v2, the pre-trained model (VideoMAE$_{PT}$), and a model that has been further fine-tuned for action recognition on the SSv2 dataset (Goyal et al., 2017) (VideoMAE-SSV2). For VideoMAE-PT, we pool over patches to calculate the embeddings, and use the negative $L_2$ loss on the decoded patches as the single-video metric. For VideoMAE-SSV2, we follow Ge et al. (2024) and choose the penultimate layer before classification for the embeddings, and the negative entropy over action class predictions as the single-video metric.

**MooG** MooG (van Steenkiste et al., 2024) is a recurrent model trained for next-frame prediction. It operates by maintaining and updating an internal state constituting of off-the-grid latent tokens, which can be decoded to predict the next frame. These latents are first randomly intialized, and then updated on each iteration by a transformer model which cross attends to the image features of the corresponding frame, followed by a set of self-attention layers. The

decoder converts the latent state back to pixels by querying the latents through cross-attention with fixed grid-based features (Sajjadi et al., 2022; Jaegle et al., 2022). We slightly modify the original model by decoding the next-frame from the "corrected" state to simplify the implementation. van Steenkiste et al. (2024) showed how MooG learns representations that encode both appearance and motion, as they can be decoded into both reconstructed video and point-tracks using shallow decoder heads. However, due to the size of the latent space, we will only consider its prediction error in the form of PSNR as a single-video metric.

## 4. Human evaluation

To evaluate whether the proposed metrics are useful, we compare them to human judgements. We propose to source fine-grained human annotations for generated videos, focusing on the following dimensions. *Consistency:* Whether the appearance and motion of objects and background appear consistent over time; *Interactions:* Whether the interactions between the objects are realistic (if any take place); *Realism:* Whether the objects and background are depicted realistically; and *Speed:* Whether the motion of the objects and camera is perceived as being slow, normal, or fast.

Together, these are intended to capture common failure modes of generative video models, such as objects changing in shape unnaturally, discrete jumps or jitter, poor interactions, such as objects that blend when they come together, or highly implausible scenes. We allow humans to provide real-valued judgements where possible to reflect that a generated video might typically meet the criteria above only to a variable degree. We refer to Appendix B.3 for details about questions, including an overview of the evaluation UI.

## 5. Results

We evaluate TRAJAN and alternative approaches in three different scenarios for how one might make use of them as metrics in practice: conducting per-video (§5.1) quality assessments, comparing generated to real videos (§5.2),

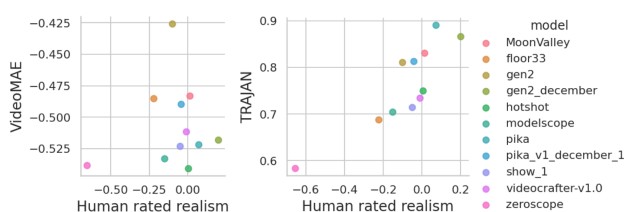

*Figure 3.* TRAJAN (right) captures the rank ordering of human preferences for different generative video models very well relative to VideoMAE (left). When correlated with human realism ratings, TRAJAN achieves a spearman rank coefficient of 0.9, while VideoMAE only achieves 0.07.

| Method | EvalCrafter | | | | VideoPhy | | | |
| --- | --- | --- | --- | --- | --- | --- | --- | --- |
| | Motion Consist. | App. Consist. | Realism | Interacts | Motion Consist. | App. Consist. | Realism | Interacts |
| V-MAE$_{PT}$ - $L_2$ | -0.00 | 0.00 | 0.04 | -0.05 | -0.03 | -0.06 | 0.06 | 0.02 |
| I3D - Entropy | -0.01 | -0.03 | -0.02 | 0.03 | 0.09 | 0.08 | 0.14 | **0.09** |
| RAFT - Warp | **0.28** | 0.27 | 0.25 | 0.13 | 0.20 | 0.26 | 0.18 | 0.03 |
| MooG - PSNR | 0.21 | 0.19 | 0.17 | 0.05 | 0.11 | 0.16 | 0.07 | 0.01 |
| TRAJAN | **0.29** | **0.29** | **0.27** | **0.19** | **0.25** | **0.32** | **0.29** | **0.09** |
| Inter-rater $\sigma$ | 0.49 | 0.46 | 0.47 | 0.53 | 0.48 | 0.46 | 0.48 | 0.53 |

*Table 1.* Spearman's rank coefficients between human ratings and automated metrics for a subset of videos from EvalCrafter (Liu et al., 2024c) and VideoPhy (Bansal et al., 2024b) (higher is better). Inter-rater $\sigma$ is the standard deviation of human responses (lower is better).

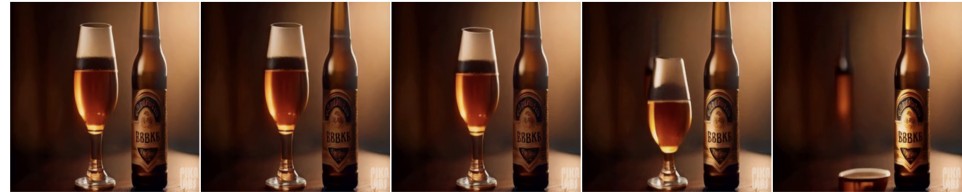

*Figure 4.* Humans rate this video to be poor in terms of all consistency / realism questions. All automated metrics score this positively. In this case, none of the metrics are capturing the unexpected disappearance and morphing of the glass.

and comparing sets of videos at a distributional-level (§5.3). Across all settings, we find that TRAJAN consistently performs better than the alternatives in its sensitivity to motion artefacts. We report additional results in Appendix A and complete experimental details in Appendix B.

### 5.1. Assessing generated videos individually

In many cases, we want a metric that can be used to determine the quality of the motion for a single video, i.e. without a reference. It is increasingly common to encounter generative video models for which we do not have access to training data, or that are too computationally expensive to sample many videos from. Here we investigate metrics to evaluate different aspects of motion on a per-video basis. We source generated videos from the "solid-solid" split of VideoPhy (Bansal et al., 2024b), which contains videos generated by 8 different models, and EvalCrafter (Liu et al., 2024c), using 11 different models. Although these datasets include human evaluations, they focus on aggregate metrics such as "motion quality" and do not contain enough information to evaluate consistency between raters. Therefore, in addition to the original ratings, we conduct our own human evaluation focusing on individual motion dimensions as outlined in §4. Each video is rated by 3 raters sampled randomly from a total pool of 10. We then z-score each rater's responses to account for different raters using the sliding scale differently (Appendix B.3).

A summary of our main findings are shown in Table 1 for consistency related dimensions, and Table 2 for speed re-

| Method | EvalCrafter | | VideoPhy | |
| --- | --- | --- | --- | --- |
| | Camera speed | Object speed | Camera speed | Object speed |
| RAFT - Mag. | 0.32 | **0.57** | 0.12 | **0.63** |
| TRAJAN- Len. | 0.26 | **0.58** | 0.16 | 0.47 |
| TRAJAN- Radii | **0.42** | 0.01 | **0.39** | -0.28 |
| Inter-rater $\sigma$ | 0.06 | 0.08 | 0.07 | 0.08 |

*Table 2.* Spearman's rank coefficients between human ratings and automated metrics for the *amount of motion* in generated videos from the EvalCrafter and VideoPhy datasets.

lated dimensions. We also show overall correlations for TRAJAN and VideoMAE against human-rated realism for the 11 different generative video models from EvalCrafter in Figure 3. All numbers are reported as Spearman's rank coefficients. We find interesting and surprising results from the human study. First, the humans themselves are not consistent with each other. We do not have sufficient numbers of overlapping examples for rater pairs, so we instead compute the inter-rater standard deviation $\sigma$ for each video for each question type and average across all videos in the dataset. Larger $\sigma$ therefore indicates more discrepancy between raters for a given question type. For example, given that ratings were z-scored by participant, the standard deviation for a single rater *across the dataset* would be 1.0. A $\sigma$ of 0.5 therefore indicates that there is approximately half as much variation in rater responses to a single question as there is variation in a single rater across the dataset.

Across all question types, in both datasets, TRAJAN best correlates with human judgements, and performs similar

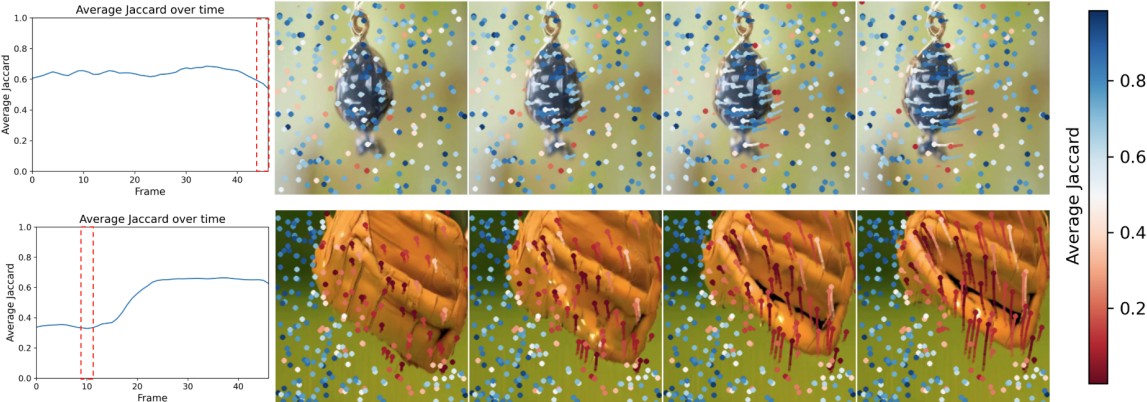

*Figure 5.* The Average Jaccard (reconstruction quality – higher AJ means higher reconstruction quality) from TRAJAN can be used to detect generated video inconsistencies at specific points in space and at specific moments in time. Here we show two examples (top and bottom). On the **left** the AJ is averaged over all points for each frame of the video. This highlights moments in time where there are temporal inconsistencies overall. On the **right** we visualize the AJ for each visible point track for 4 consecutive frames centered on the frame having the lowest AJ score (indicated with the dashed lines on the left). Red indicates poor reconstruction (low AJ) while blue indicates good reconstruction (high AJ). In the top case, the motion is consistent over most of space, so there is overall higher AJ across the points. In the bottom case, the glove's appearance morphs over the frames, leading to worse reconstruction for points located on and immediately around the glove.

to RAFT for consistency-related categories. While this may be expected for motion and appearance consistency, TRAJAN also correlates best for *realism* both for the video overall, and for the object interactions. This holds not just on a per-video basis, but also tracks which generative video model humans prefer overall for realism (Figure 3). If we condition on only high motion videos (those rated by humans as having camera speeds greater than or equal to "medium"), where we expect motion-based metrics to be most informative, we see even stronger correlations (and lower inter-rater $\sigma$s), especially for the EvalCrafter dataset (Table 3 and Appendix A). The point tracks can also be used to measure motion *amount* (Table 2), with both point track lengths and optical flow magnitude being predictive of object speeds, and point track radii (see Appendix B.2.1 for details on how these are calculated) being predictive of camera speed.

However, even TRAJAN is not a perfect predictor of human judgements. For example, in Figure 4, a beer glass disappears by collapsing into the table. All metrics rate this video highly, as the motion is smooth and easily tracked. However, this is a very unrealistic motion, since there is nothing that would cause the glass to collapse. Future work would be needed to develop metrics with expectations about what *should* happen rather than simply tracking what *does* happen.

**Original EvalCrafter dataset** To ensure that our metrics are not biased to the human evaluations we collected, we also compare to the human ratings released on the EvalCrafter dataset by the original authors in Appendix A. How-

| Method | Consist. | Quality | Visual | T2V | Subj. |
|---|---|---|---|---|---|
| V-MAE$_{PT}$ | -0.19 | -0.16 | -0.06 | -0.07 | -0.02 |
| I3D | 0.09 | **0.24** | 0.22 | **0.25** | 0.14 |
| RAFT | 0.08 | -0.01 | 0.09 | 0.13 | 0.15 |
| MooG | 0.13 | 0.08 | 0.01 | 0.11 | 0.03 |
| TRAJAN | **0.33** | **0.24** | **0.28** | **0.25** | **0.23** |

*Table 3.* Spearman rank coefficients on the original EvalCrafter dataset for their categories of temporal consistency, motion quality, visual quality, text-to-video similarity, and subjective likeness, when holding the overall amount of motion fixed.

ever, we found that human judgements in this dataset (across temporal consistency, visual quality, text-to-video similarity, and subjective likeness), were mostly driven by simple measures of overall motion (i.e. videos with significant motion are worse, regardless of how realistic that motion appears). We therefore re-analyze the data controlling for overall motion by binning videos based on their optical flow magnitude (RAFT - Mag), and then evaluate Spearman rank coefficients within bins falling into the top half of the dataset (those where there was significant motion). We average the coefficients across bins. Here, TRAJAN strongly outperforms other metrics in explaining temporal and visual consistency, as well as subjective likeness, suggesting that it captures more than just overall motion magnitudes in explaining motion quality.

**Original VideoPhy dataset** As a further external validation, we report the area under the ROC curve using the original human labels for physical consistency contained in the VideoPhy dataset (Figure 6). RAFT, MooG and TRAJAN are the top-performing models in this case. We also

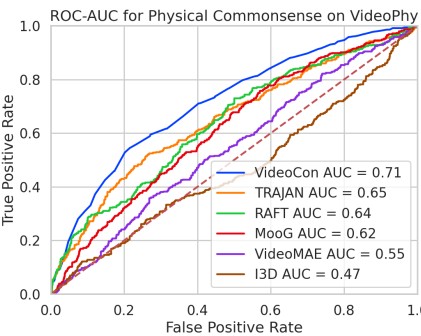

*Figure 6.* ROC-AUC Scores for physical consistency on Video-Phy dataset. Note that VideoCon is *fine-tuned* on human labels, whereas all other methods are not.

compare to VideoCon as proposed in Bansal et al. (2024b) as a method for evaluating physical consistency in generated videos, which fine-tunes a vision-language model on a subset of the human labels. Without any fine-tuning on human data, TRAJAN performs strikingly well, which indicates its generality for evaluating motion in videos originating from different models and datasets (i.e. prompts).

**Spatiotemporal error localization with TRAJAN** Finally, we demonstrate how TRAJAN can be used to localize generative video errors in both space and time for an individual video. In particular, TRAJAN allows us to calculate reconstruction quality (through the Average Jaccard metric applied to the reconstructed points) for *each point in space* over *short windows in time* to determine *where* and *when* motion and appearance inconsistencies occur.

Figure 5 shows a generated video which TRAJAN scores as highly consistent (top) and highly inconsistent (bottom). Looking at the progression over time on the left, the consistent video maintains reasonably high Average Jaccard. However, for the video at the bottom, where the glove's appearance morphs throughout the beginning of the video before stabilizing, we see a dramatic change in Average Jaccard. Visualizing the spatial errors around these poorly reconstructed frames (on the right) shows that the points on the glove are contributing the most to the low score.

### 5.2. Comparing real and generated video pairs

Many video models are evaluated by conditioning on a few frames of a *real* video, and then predicting how that video should unfold in the future (Whitney et al., 2023; Wang et al., 2017; Lin et al., 2020). However, comparing future generated and real frames is not straightforward. Comparisons in pixel space can suffer from inherently uncertain futures such as objects that might enter the frame, or panning a camera such that new parts of the scene are visible (Figure 7a). In this section, we investigate whether comparisons in the latent space of various models can capture

| Method | PSNR | SSIM |
|---|---|---|
| V-MAE$_{PT}$ | 0.35 | 0.53 |
| I3D | **0.24** | 0.26 |
| TRAJAN | **0.23** | **0.13** |

*Table 4.* Spearman's $\rho$ between latent space distances and PSNR / SSIM scores for high motion videos generated by WALT.

video-to-video similarity better than pixel-based scores.

To do so, we take the WALT video diffusion model (Gupta et al., 2024) and train it on the Kinetics-600 dataset (Carreira et al., 2018) to predict future video frames conditioned on 2 real latent frames. We sample checkpoints from throughout training, and calculate video embeddings using TRAJAN, VideoMAE and I3D on 2364 generated and corresponding real samples with high motion as estimated by the length of the point trajectories.

In Table 4, we show that the distances in model latent space are not well correlated with either PSNR or SSIM for any of the models, and the correlation is lowest for TRAJAN. This suggests that the TRAJAN latent space is capturing something fundamentally different from pixel error, which we highlight with examples in Figure 7. In particular, TRA-JAN is sensitive to differences in *motion* but not appearance, while distances in the VideoMAE and I3D latent spaces are more sensitive to overall recognizability. This suggests that the TRAJAN latent space is useful for measuring similarities in the motion between videos, even when the exact pixels or overall appearance are not a perfect match.

### 5.3. Comparing video distributions

Finally, we demonstrate how to compare empirical distributions of videos, which is the usual paradigm for metrics based on FVD (Unterthiner et al., 2018). We obtain real videos from the UCF101 dataset, which consists of 13,320 videos recorded in the wild that show humans performing different types of actions (Soomro et al., 2012), and synthetically corrupt them using the 5 levels of "elastic transformations" from Ge et al. (2024). Corruption levels 1.1 and 1.2 apply low frequency deformations, while 2.1, 2.2, and 2.3 apply high frequency deformations (Figures 13 & 14 in Appendix B)[1]. There are two corruption modes: a spatiotemporal (ST) mode, where video frames are distorted independently using different parameters, and a spatial (S) mode where the same set of parameters is used to distort all the frames. Ge et al. (2024) propose to use the *ratio* of these modes to measure how sensitive metrics are to temporal corruptions.

We first apply the standard technique of computing the Fréchet distance between the latent representations com-

---

[1]Parameters were obtained via personal correspondence.

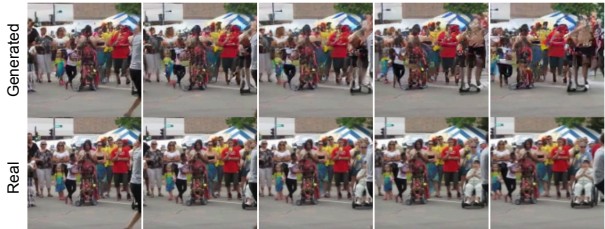
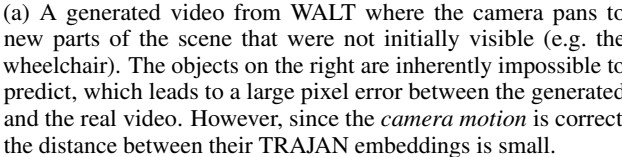
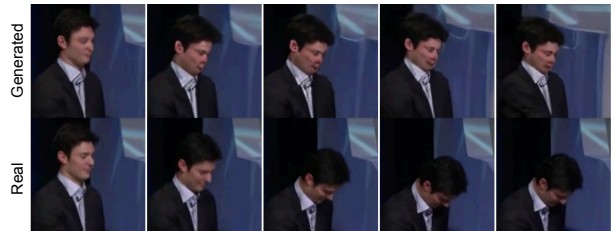

(a) A generated video from WALT where the camera pans to new parts of the scene that were not initially visible (e.g. the wheelchair). The objects on the right are inherently impossible to predict, which leads to a large pixel error between the generated and the real video. However, since the *camera motion* is correct, the distance between their TRAJAN embeddings is small.

(b) A generated video from WALT with incorrect predicted motion. Despite the incorrect motion, the I3D and VideoMAE latent space distance between the generated and real videos is very low (rank 38, 74 / 2364), while the distance in the TRAJAN embeddings is high (rank 1655 / 2364).

Figure 7. WALT extrapolations for two videos (top: generated, bottom: real).

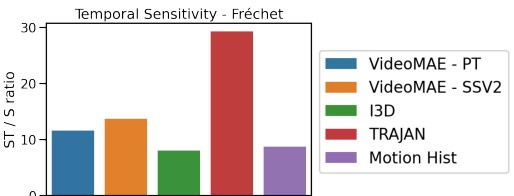
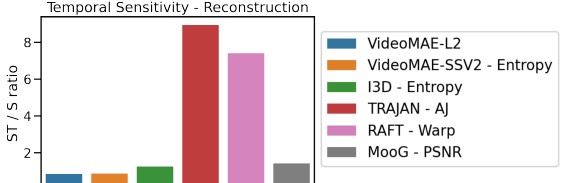

Figure 8. Comparing different methods for detecting spatiotemporal distortions (ST) relative to spatial-only distortions (S) on the UCF-101 dataset (Ge et al., 2024) in terms of (left) Fréchet distances in latent space and (right) per-video ordinal scores. TRAJAN and RAFT warp error are particularly sensitive to spatiotemporal distortions (high ST / S ratios), while motion histograms (Liu et al., 2024a) and appearance-based methods are equally sensitive to spatiotemporal vs. spatial-only distortions. See Figure 13 and Figure 14 for examples.

puted for videos within each set, using TRAJAN and four of the models presented in §3.2. Figure 8 (left) reports the average temporal sensitivity across different levels of corruption strength as in Ge et al. (2024). TRAJAN is the most sensitive to temporal distortions.

To bridge distribution level and single-video metrics, we compute the latter for each video and take the average across the set (Figure 8, right). TRAJAN and the RAFT warp error are particularly sensitive to temporal distortions, while motion histograms (Liu et al., 2024a) and appearance-based methods are not. Although distributional comparisons yield the highest overall sensitivity, the strong performance of TRAJAN suggests that per-video metrics based on motion can be used as a replacement for distribution-based metrics such as FVD if a reference dataset is not available or if sample sizes are limited.

## 6. Conclusion

We investigated different methods for evaluating motion in generated videos, and proposed TRAJAN, a novel architecture for auto-encoding dense point tracks. We showed that TRAJAN's reconstruction error correlates remarkably well with human judgements of motion consistency, appearance consistency, realism, and even visual quality, and can be used to localize spatiotemporal errors in generated video. TRAJAN can also be used to compare motions of generated

and real video pairs even when the appearance is not maintained. Finally, TRAJAN presents an alternative to FVD for measuring generated video quality, as it is significantly more sensitive to motion artefacts at the distribution level. Together, these results suggest an alternative approach to evaluating generated videos: not just by using a different backbone for FVD, but by moving away from costly distributional comparisons all together, and focusing on comparing generated / real video pairs in latent space or evaluating individual videos (all three are supported by TRAJAN).

Despite the strengths of TRAJAN, we also found that there are shortcomings. By conducting a detailed human study, we showed that while TRAJAN predicted human judgements better than a range of alternative motion- and appearance-based metrics, there are videos for which no automated metric captures human ratings. To capture human judgements in these scenarios, future work will need to develop more advanced models that can make more accurate judgements not just about how something *does* move, but how it *should* move if it was obeying physical principles.

Furthermore, individual people seem to care about different aspects of videos for evaluating basic properties like temporal and appearance consistency, leading to low consistency in evaluations across human participants. This suggests that future work is needed to determine how to elicit better judgements from human raters, and suggests caution for using humans as a gold standard for adjudicating between models.

## Acknowledgements

We would like to thank Mike Mozer for helpful comments and suggestions to improve our paper, and Viorica Patraucean, Yiwen Luo, Lily Pagan, and Nishita Shetty for help preparing the human study. We would like to thank Songwei Ge and Ge Ya Luo for sharing details about the video corruptions used in their work.

## Impact Statement

This paper presents work whose goal is to advance the field of Machine Learning. By presenting a novel evaluation framework for generated videos, this work has the potential to aid in the detection of generated video content. It also has the potential to allow generative video model designers to better understand whether their models will appear realistic to human observers. However, we do not feel that these risks outweigh the potential benefits. There are many other potential societal consequences of our work, but we do not feel that they must be specifically highlighted here.

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

# A. Additional Results

## A.1. MMD with different backbones

An alternative to using Fréchet distance for comparing embeddings is to make us of the Maximum Mean Discrepancy (Gretton et al., 2012) as a distance function. This idea was first proposed for comparing distributions of images in Bińkowski et al. (2018), and briefly explored for video in Unterthiner et al. (2018). MMD is a kernel-based approach to comparing distributions without assuming a particular form (such as a Gaussianity when using Fréchet distance). An unbiased estimator of the squared MMD is:

$$\sum_{i \neq j}^{m} \frac{k(x_i, x_j)}{m(m-1)} - 2 \sum_{i}^{m} \sum_{j}^{n} \frac{k(x_i, y_j)}{mn} + \sum_{i \neq j}^{n} \frac{k(y_i, y_j)}{n(n-1)}. \tag{1}$$

where $\{x_i\}_i^m$ and $\{y_j\}_j^n$ are samples drawn from the respective distributions we wish to compare, and $k(\cdot, \cdot)$ is a kernel function. Here we use the polynomial kernel function $k(a, b) := (a^T b + 1)^3$ as in Unterthiner et al. (2018).

In Figure 9 we report the results of using MMD instead of the Fréchet distance for calculating the sensitivity of different models to temporal corruptions on the UCF-101 dataset. Defined as in (1), the MMD can occasionally be negative. If the MMD is negative (which was the case for the motion histograms model for several corruption levels), we replace it with $1e - 6$. With MMD, all models are significantly more sensitive to temporal corruptions, with motion histograms showing the largest sensitivity. Upon further investigation we noticed that this is primarily due to 3 cases where the spatial-only corruptions gave negative MMD scores.

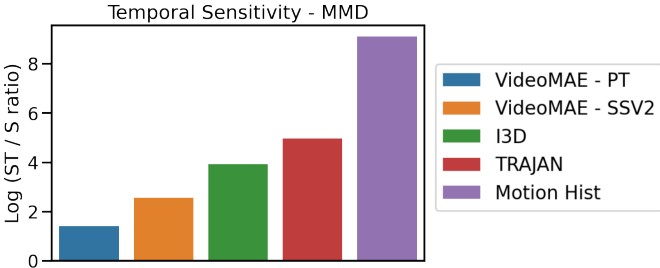

*Figure 9.* MMD for UCF-101 corruptions

## A.2. Metric changes during training

We measure how each of the different metrics tracks WALT's (Gupta et al., 2024) training progress at 6 different checkpoints in training (Figure 10). For the distribution-level metrics, we see similar performance using TRAJAN, VideoMAE or I3D. For the single-video metrics, only TRAJAN tracks WALT's training reasonably well. However, most of the change in all metrics is driven by the first 100k steps, after which all metrics become more-or-less constant.

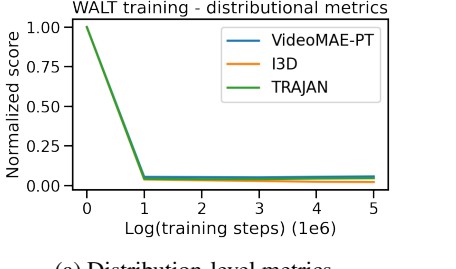

(a) Distribution-level metrics.

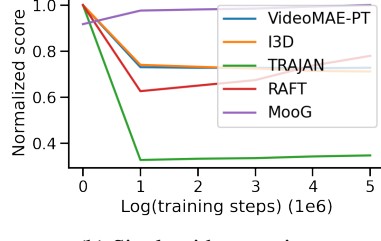

(b) Single-video metrics.

*Figure 10.* Metrics over training iterations for WALT.

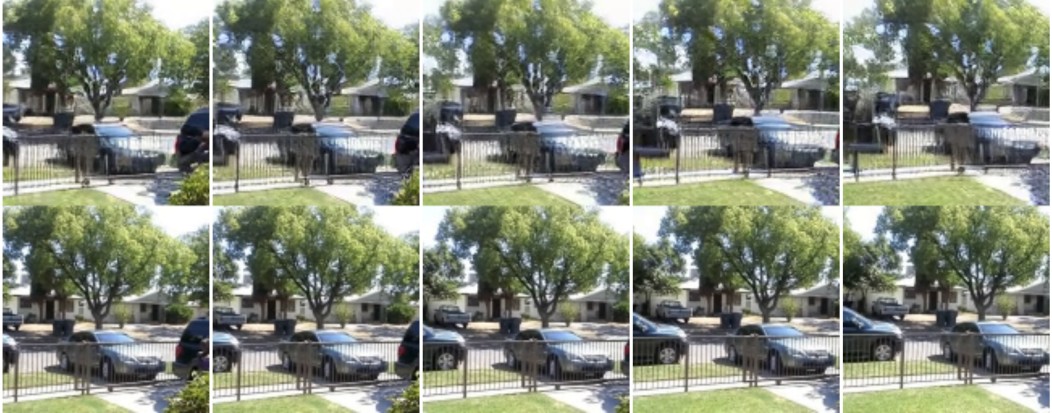

(a) Generated (top) with real (bottom) from WALT. In this example, the camera pans to the left. The pixel error between these two videos is very high, since as the camera moves to the left, it does not fill in the left side of the image successfully, nor does it predict the exact spacing of the grating. However, since the motion of all the objects is correct, the distance between these videos in the TRAJAN feature space is low.

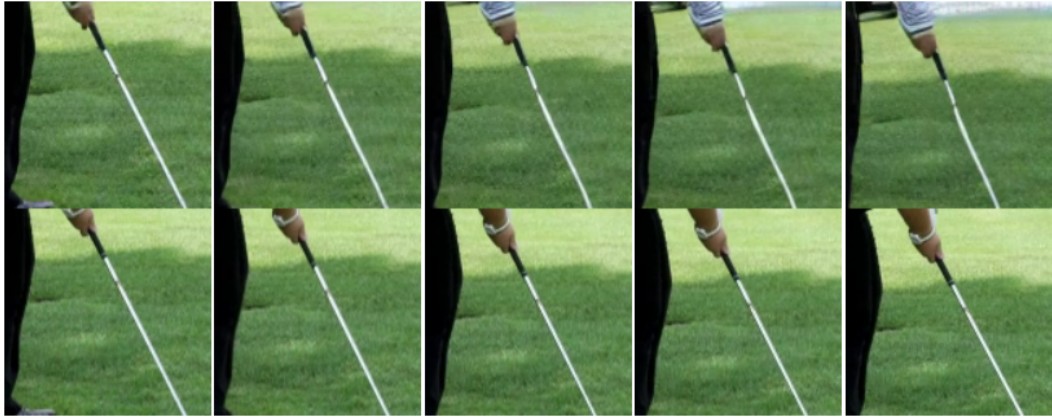

(b) Generated (top) with real (bottom) from WALT. In this example, the golfer's arm moves into the frame from the top. The pixel error between these two videos is high, because the color of the man's jacket is incorrect, and there is a generated background object at the top of the video that does not actually exist. However, the distance in the TRAJAN feature space is low, because the motion of the golfer's arm is correct.

*Figure 11.* Additional paired WALT generations.

### A.3. Additional WALT examples for generated vs. real videos

In Figure 11 we show additional examples of comparing real vs. generated videos from the WALT model (Gupta et al., 2024) which have different amounts of average pixel error. In Figure 11a, there is very high pixel error because of objects that could not be seen as the camera pans to the left, and in Figure 11b, the pixel error is reasonably high because the predicted color of the golfer's jacket is incorrect, and an extra background object is generated which does not exist. In both cases, the distance between the real and generated video in TRAJAN feature space is low, because the motion of the camera (in the first case), or the man (in the second case) is correct.

### A.4. Additional results on VideoPhy (Bansal et al., 2024b) and EvalCrafter (Liu et al., 2024c)

Table 5 shows the results of different automated metrics on the full EvalCrafter (Liu et al., 2024c) dataset using their labels. We found that across the dataset, RAFT - Mag, the average optical flow magnitude across the video, is a very good predictor of human judgements in all conditions. This suggests that, on average, videos that are faster are more poorly rated by humans generally.

| Method | Consistency | Visual | T2V | Subj. |
|---|---|---|---|---|
| VideoMAE$_{PT}$ | -0.19 | -0.12 | -0.05 | -0.09 |
| I3D | -0.00 | 0.01 | 0.00 | -0.01 |
| RAFT - Warp | **0.61** | **0.42** | 0.18 | **0.34** |
| RAFT - Mag | -0.58 | -0.37 | -0.17 | -0.32 |
| MooG | 0.42 | 0.23 | 0.10 | 0.22 |
| TRAJAN | 0.52 | 0.36 | **0.20** | 0.32 |

*Table 5.* Results on full EvalCrafter (Liu et al., 2024c) dataset.

| | EvalCrafter | | | VideoPhy | | |
|---|---|---|---|---|---|---|
| | **Motion Consistency** | **Appearance Consistency** | **Realism** | **Motion Consistency** | **Appearance Consistency** | **Realism** |
| VideoMAE$_{PT}$ | 0.06 | -0.05 | 0.01 | -0.01 | 0.04 | 0.01 |
| I3D | -0.05 | 0.08 | -0.04 | 0.14 | 0.09 | **0.19** |
| RAFT | 0.21 | 0.19 | 0.11 | 0.30 | **0.40** | 0.09 |
| MooG | 0.10 | 0.08 | -0.01 | 0.19 | 0.31 | 0.13 |
| TRAJAN | **0.44** | **0.51** | **0.34** | 0.33 | **0.41** | 0.14 |
| Rater $\sigma$ | 0.43 | 0.41 | 0.41 | 0.47 | 0.46 | 0.44 |

*Table 6.* Spearman's rank coefficients between human ratings and automated metrics for the medium - high motion subset of the data. In this subset, people are more consistent in their scores (lower rater $\sigma$), and in most cases, TRAJAN also better predicts human ratings. In the case of VideoPhy realism, there are specific failure modes in the dataset where some models produce videos with inconsistent motion but realistic individual frames, see Figure 12a, but humans still rate these as realistic.

Table 6 gives results for both EvalCrafter and VideoPhy when looking only at medium and high motion samples (those with human ratings of "medium" or "high" on the camera motion question). In these cases, TRAJAN provides better correlations to human ratings, and humans are more consistent with each other (lower inter-rater $\sigma$s). This suggests, similar to the results on the external human datasets for EvalCrafter, that TRAJAN is capturing human ratings of consistency beyond just overall amount of motion.

We also provide further qualitative examples of how people rate videos from the VideoPhy (Bansal et al., 2024b) and EvalCrafter (Liu et al., 2024c) datasets in our human study in Figure 12. In Figure 12a, people rate the video as realistic, despite the fact that the video is temporally inconsistent. In Figure 12b, people rate the video as unrealistic and inconsistent, even though the motion of the dogs playing poker looks reasonable. In Figure 12c, people also rate the video as both unrealistic and inconsistent in appearance, even though the appearance does not change. People therefore seem strongly affected by the semantics of the objects even when prompted to ignore this information.

**Validation Study.** We validate the UI for our human study by conducting a second human study on the same subset of videos obtained from the EvalCrafter dataset. For this study, we replaced the slider scales with preset answer options corresponding to a 5-point Likert scale. Here option 1 implies a video that is very unrealistic (for questions about realism) or inconsistent (for questions about consistency) along this dimension, and option 5 implies a video that is very realistic (for questions about realism) or very consistent (for questions about consistency). For this study we used 5 raters per question.

Results are reported in Table 7, where we post-processes in the same way as was done for Table 1. To make the results more comparable, we mapped the 1-5 Likert scale onto the slider scale by interpreting each scale as increments of 20. It can be seen how TRAJAN and RAFT best correlate human judgement similar to in Table 1 (left) for consistency-related categories and realism, while TRAJAN performs clearly better for object interactions.

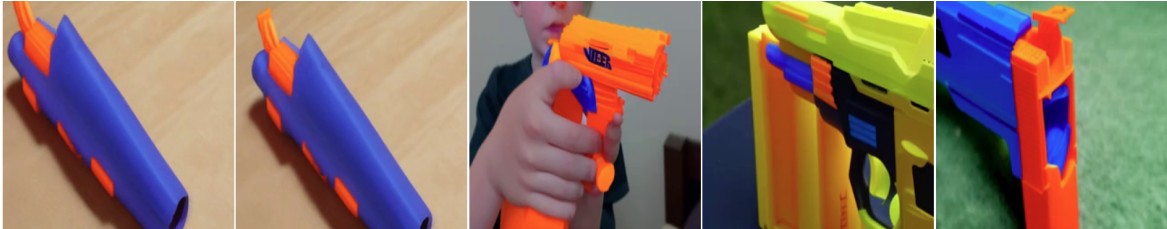

(a) Example from the VideoPhy dataset sampled every 5 frames. This sample is scored in the upper half of the dataset for realism and appearance consistency by humans, and receives a physical consistency score of 1 from the original VideoPhy dataset.

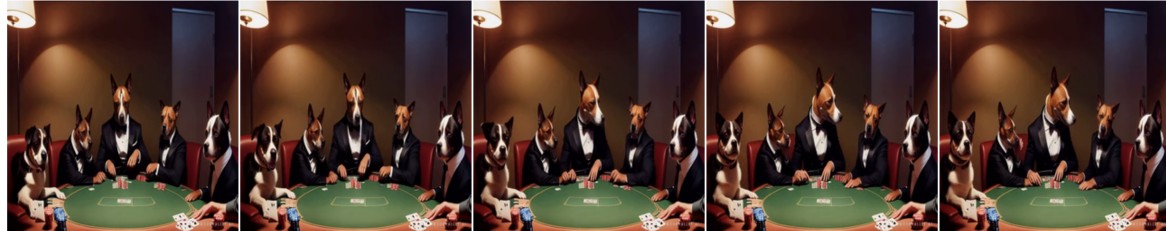

(b) Humans rate badly on all consistency / realism metrics. All automated metrics score this highly. In this case, motion consistency and appearance consistency qualitatively appear good, but people seem unable to separate realism from consistency.

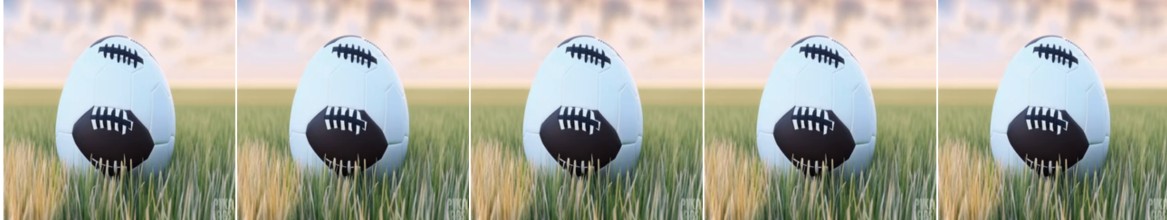

(c) Humans rate badly on realism metrics. All automated metrics score this highly. In this case, motion consistency and appearance consistency qualitatively appear good, but people seem unable to separate the realism of the soccer / football from consistency.

*Figure 12.* Examples from VideoPhy and EvalCrafter with human ratings.

| Method | Motion Consistency | Appearance Consistency | Realism | Interacts |
|---|---|---|---|---|
| V-MAE$_{PT}$ - $L_2$ | 0.04 | 0.07 | 0.10 | -0.07 |
| I3D - Entropy | -0.05 | -0.06 | -0.02 | 0.02 |
| RAFT - Warp | **0.39** | **0.43** | **0.29** | 0.18 |
| MooG - PSNR | 0.28 | 0.32 | 0.26 | 0.09 |
| TRAJAN | **0.38** | 0.40 | 0.26 | **0.26** |
| Inter-rater $\sigma$ | 0.09 | 0.07 | 0.09 | 0.12 |

*Table 7.* **Validation Study.** Spearman's rank coefficients between human ratings and automated metrics for a subset of videos from EvalCrafter (Liu et al., 2024c) (higher is better). Inter-rater $\sigma$ is the standard deviation of human responses (lower is better).

## B. Experiment Details

### B.1. Datasets

#### B.1.1. UCF101

For our experiments with synthethic distortions, we make use of the UCF101 dataset (Soomro et al., 2012), which consists of 13,320 videos recorded in the wild that show humans performing different types of actions. We make use of the version available at `https://www.tensorflow.org/datasets/catalog/ucf101` having $256 \times 256$ resolution. Similar to prior work (Ge et al., 2024), we obtain the 'ground-truth' reference distribution by combining the first 32 frames of videos in the train and test split.

To obtain distorted videos, we consider the *elastic transformation* from Ge et al. (2024). Details were obtained via personal correspondence with the authors, which we reproduce in the following.

**Elastic Transformation (Ge et al., 2024)** The reference implementation for this transformation is `https://github.com/hendrycks/robustness/blob/master/ImageNet-C/imagenet_c/imagenet_c/corruptions.py`, which works by first performing an affine- and then an elastic transformation. The reference implementation suggests five levels of degradation (Hendrycks & Dietterich, 2018). The first two levels (referenced as 1.1 and 1.2 in **??**) primarily trade-off the strength of the affine transformation with that of the elastic transform, such that 1.2 creates more global distortions to the shape of objects. Levels 2.1, 2.2, and 2.3 fix the affine transformation at small intensity, and only increase the strength of the elastic part. This results in much more local distortions associated with high-frequency noise. Ge et al. (2024) adopt these sample levels, except that they adjust the resolution parameter from 244 to 128. Here we adopt the same distortion levels as in Ge et al. (2024).

To compute the temporal sensitivity of a metric, Ge et al. (2024) proposes to evaluate them on both *spatially* distorted videos and on *spatiotemporally* distorted videos, and report the ratio of the two. In practice, the elastic transformation is applied at a per-frame level, where the distortion levels parametrize a distribution of corruptions that are sampled from. When using the same seed between frames in a video, the same corruption is applied to each frame, creating only a spatial effect. To obtain a spatiotemporal effect, a different corruption is sampled for each frame in the video.

In Figures 13 & 14 we show an example of a UCF-101 video with spatial- or spatio-temporal corruptions applied using the five levels of elastic transformation from Ge et al. (2024). Note that in the spatial setting, the same corruption is applied to each frame, while in the spatiotemporal setting the corruption parameters are resampled for each frame.

#### B.1.2. VIDEOPHY

We make use of the "solid-solid" split of VideoPhy (Bansal et al., 2024b) available for download at `https://huggingface.co/datasets/videophysics/videophy_test_public`. These include generated videos from several competitive models, including CogVideoX (2b and 5b) (Yang et al., 2024), Gen-2 (Esser et al., 2023), LaVIE (Wang et al., 2023c), OpenSora (Zheng et al.), Pika (Pika), SVD (Blattmann et al., 2023a), VideoCrafter2 (Chen et al., 2024), and ZeroScope (Sterling). Note that at the time of preparing this submission, we were unable to download videos contained in the dataset belonging to Luma Dream Machine, which were made available at a later date. We pre-processed all videos to $256 \times 256$ resolution.

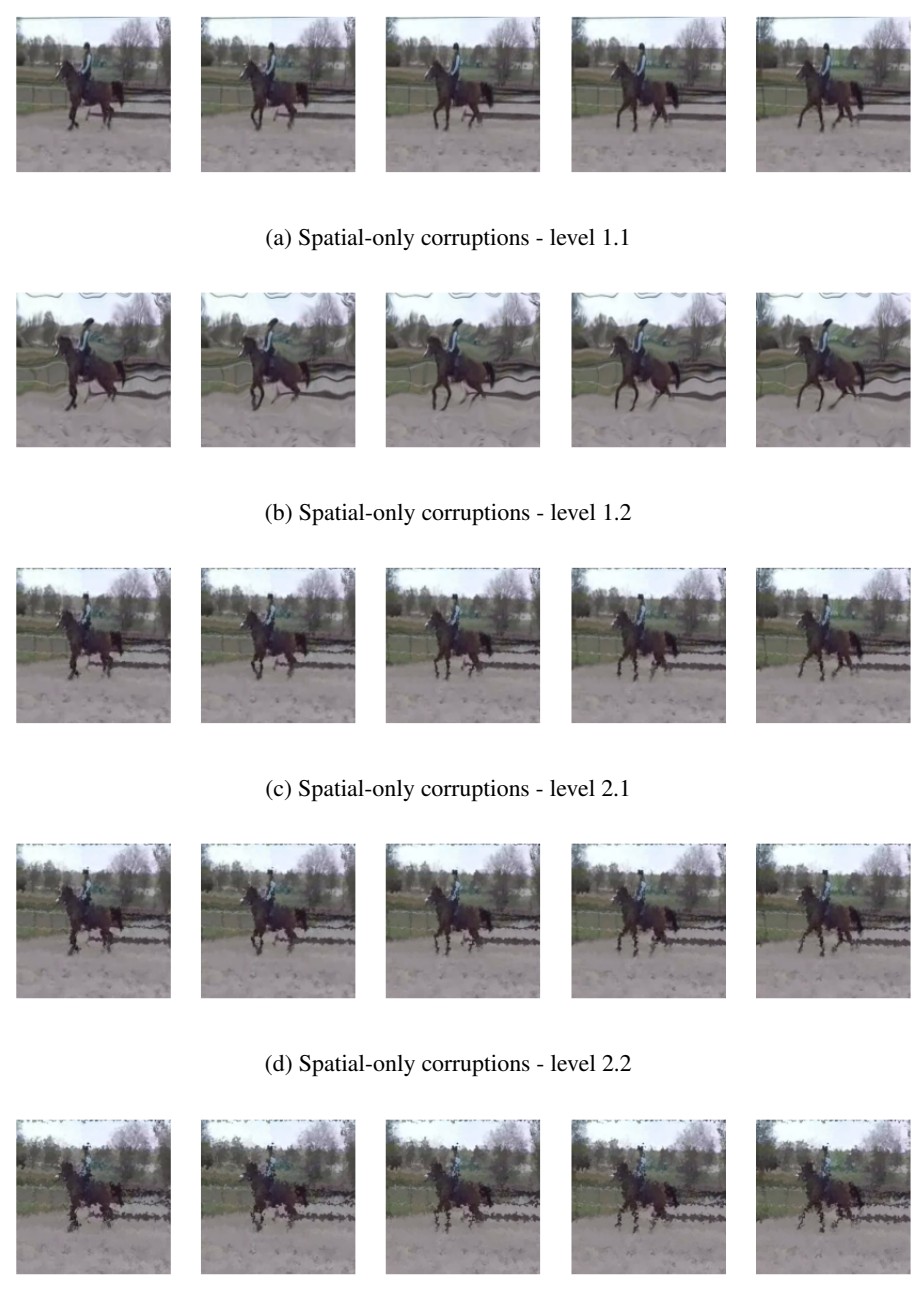

(a) Spatial-only corruptions - level 1.1

(b) Spatial-only corruptions - level 1.2

(c) Spatial-only corruptions - level 2.1

(d) Spatial-only corruptions - level 2.2

(e) Spatial-only corruptions - level 2.3

*Figure 13.* A UCF-101 video with *spatial* corruptions using the five levels of elastic transformation from Ge et al. (2024).

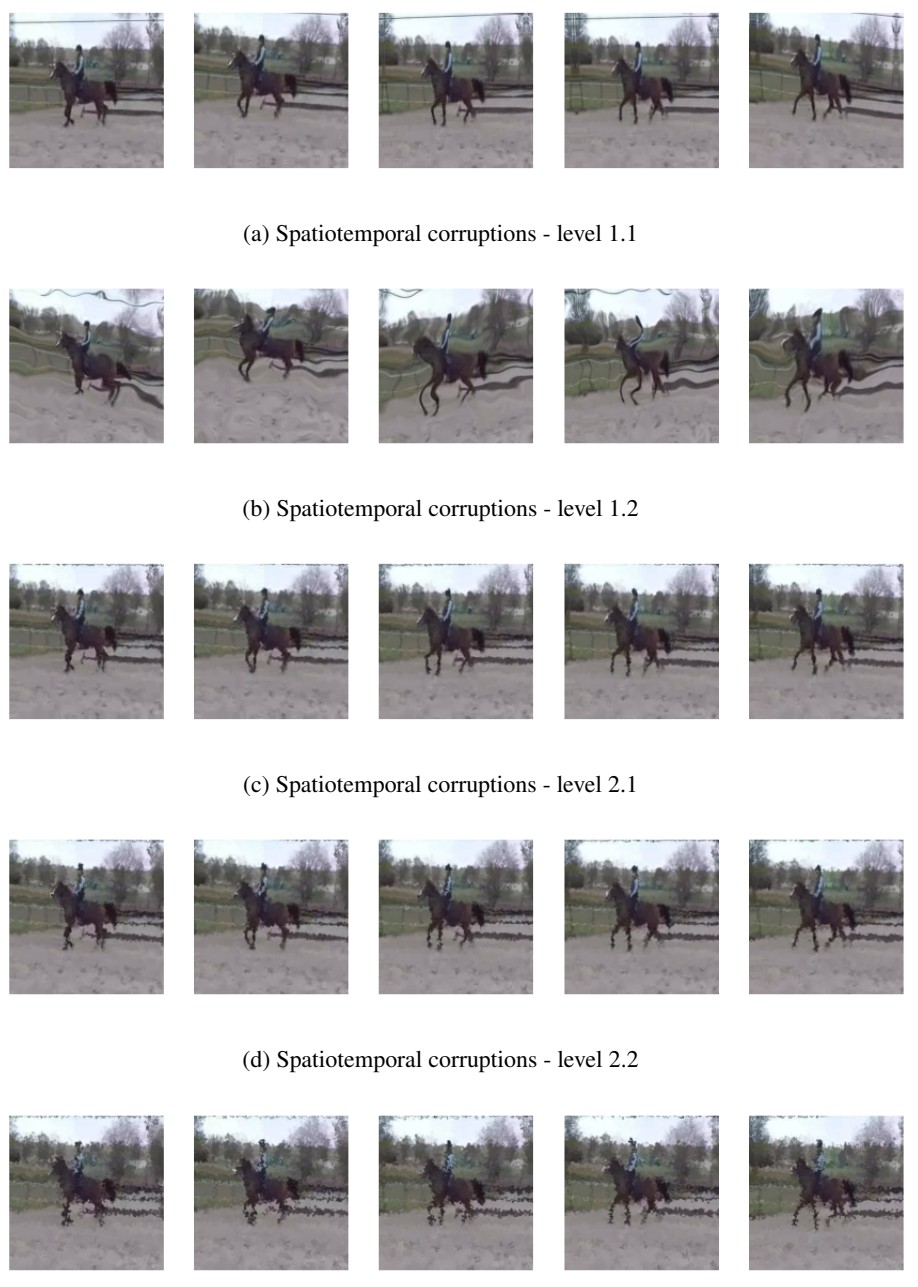

(a) Spatiotemporal corruptions - level 1.1

(b) Spatiotemporal corruptions - level 1.2

(c) Spatiotemporal corruptions - level 2.1

(d) Spatiotemporal corruptions - level 2.2

(e) Spatiotemporal corruptions - level 2.3

*Figure 14.* A UCF-101 video with *spatiotemporal* corruptions using the five levels of elastic transformation from Ge et al. (2024).

### B.1.3. EVALCRAFTER

The EvalCrafter dataset (Liu et al., 2024c) contains generated videos for a diverse set of prompts and is available for download at `https://huggingface.co/datasets/RaphaelLiu/EvalCrafter_T2V_Dataset`. To obtain a comparable size to VideoPhy, while preserving diversity as much as possible, we randomly select 104 generated videos from 11 of the text-to-video models (the original 5 from the EvalCrafter human evaluation dataset, and 6 additional models). These models are:

1. Gen2 (December) (Esser et al., 2023)

2. MoonValley (MoonValley)

3. PikaLab (December) (Pika)

4. Show-1 (Zhang et al., 2024)

5. VideoCrafter 1.0 (Chen et al., 2023)

6. Hotshot-XL (Hotshot-XL)

7. PikaLab (Pika)

8. Gen2 (Esser et al., 2023)

9. Floor33 Pictures (Pictures)

10. ZeroScope (Sterling)

11. ModelScope (Wang et al., 2023a)

We obtain 1144 videos in this way, which we preprocess to $256 \times 256$ resolution.

### B.1.4. WALT GENERATIONS

We train a WALT (Gupta et al., 2024) diffusion model (214M params) for frame-conditional video generation on the Kinetics-600 dataset (Carreira et al., 2018). The model architecture and hyperparameters are consistent with those used in the original paper. Training is conducted for 495,000 iterations with a batch size of 256, using videos at a resolution of $128 \times 128$ with 17 temporal frames. Following WALT, the model is designed to predict three future latent frames conditioned on the initial two latent frames. To evaluate performance, we compute metrics at key checkpoints, using model weights saved at steps $1, 100,000, 200,000, 300,000, 400,000$, and $495,000$. We upsample generated videos to $256 \times 256$ resolution and make use of the first 16 frames. We show the distribution level metrics and single video metrics evaluated on samples generated at these intermediate steps in Figure 10.

### B.2. Models

#### B.2.1. TRAJAN

We train TRAJAN on a video dataset of publicly accessible videos following Doersch et al. (2024), aiming for high-quality and realistic motion. We use videos tagged as lifestyle and one-shot videos, but omit those videos from categories with low visual complexity or unrealistic motions, e.g. tutorial videos, lyrics videos, and animations. We select only 60fps videos with over 200 views, without cuts or overlays. However, a key difference from prior work (Doersch et al., 2024) is that we use 150-frame clips, whereas prior work focused on 24-frame clips for bootstrapping point tracking. We sample 15 million such videos, and for each video, we choose 4096 points uniformly at random across space and time for every clip and track them using the public BootsTAPIR (Doersch et al., 2024) model.

There are three key challenges for representing a set of point tracks as a dense vector where distances are meaningful. 1) point tracks are *orderless*, in the sense that they cannot be arranged into a sequence of tokens: unlike image pixels, tracks do not occur on a grid, and there may be no single video frame where all tracks are visible. Therefore, we wish to have a permutation-invariant representation. 2) occlusions should be treated as missing data, rather than as a fundamental part of

the motion, since two motions may be very similar in the real world even if one is partially occluded. Finally, 3) although we are representing a dense signal (every point in the image corresponds to a single track), for computational reasons we only receive a finite number of samples from the underlying function. The exact points that are chosen for tracking are not important for representing motion, so we aim for our representation to be invariant to the specific chosen points.

Let $S = \{s_{t,j}\}$ be point trajectories and occlusions, where $s_{t,j} \in \mathbb{R}^3 = (x_{t,j}, y_{t,j}, o_{t,j})$ corresponds to $x$ and $y$ positions and occlusion flag $o$ at time $t$ for the $j$th track. As mentioned, we embed all $(x_{t,j}, y_{t,j}, t)$, add a readout token, and perform self-attention using $(1 - o_{t,j})$ as an attention mask, which helps us achieve point 1) above, as the transformer sees these points as 'masked' rather than some dummy value. After self attention, we discard all tokens except the "readout" token, which now provides a fixed-length $C$-channel representation of each track. We then apply a Perceiver (Jaegle et al., 2021) to encode all track tokens; we apply no additional position encoding, meaning that the entire representation is permutation invariant, following principle 1) above. Finally, we project the latent tokens to a lower dimension resulting in a fixed-size $128 \times 64$ dimensional representation $\phi_S$ of the tracks.

As mentioned, we achieve point 3) above by decoding points that are *not* included in the autoencoder input. The autoencoder knows which track to decode because we give a query point $x_q, y_q, t_q$ on any frame, and we output the track that goes through this point. Note that this means that the autoencoder can actually output truly *dense* motion information even though it receives only a finite sample as input. As stated, we up-project the tokens in $\phi_S$ to a higher dimension with an operator $U$, apply a transformer with a readout token, and apply a linear mapping to occlusion logit $o_t^q$ and $x_t^q, y_t^q$. We train them with softmax cross entropy and Huber loss, respectively, with a weight of $5000$ on the Huber loss and $1e-8$ on the cross entropy; the disparity between loss weights is due to the fact that we mostly want the representation to be mostly invariant to occlusion and focus on motion. In initial experiments, we found that setting these weights equally led to worse performance in correlating with human judgements of realism in generated videos. We find that a naive linear up-projection operator $U$ tends to result in poor temporal localization of the query point, as the model struggles to cross-attend to the correct latent tokens. We find that we can improve performance by using an upsampling operator that first linearly up-projects, and then extracts a window of each token, concatenating along the channel axis. That is, the up-projected representation for the $l$th token is $U(\phi_S^l) = concat([f(\phi_S^l), f(\phi_S^l)[\rho t : \rho t + 128])$, where $f$ is a linear projection, $[\cdot]$ represents indexing, and $\rho$ is a stride. This can be seen as specializing the motion tokens for time $t$, so the model can more easily identify what motion information is relevant to this query. We train with Adam (Kingma & Ba, 2015) with a warmup cosine learning rate schedule with 1000 warmup steps and a peak learning rate of 2e-4 for 1M steps with a batch size of 64.

Note that not all of the videos we would like to evaluate have 150 frames of motion. Therefore, we train the model to also encode shorter clips. Given motion from a 150-frame clip, for half of the examples, we (uniformly) randomly sample an 'end' frame, and mark all points after a certain length as 'occluded'. For these examples, we only apply the loss to the frames before the 'end' frame.

We evaluate reconstruction accuracy with Average Jaccard, following TAP-Vid (Doersch et al., 2022). Given a threshold $\delta$ "true positives" ($TP$) are predictions which are within $\delta$ of the ground truth. "False positives" ($FP$) are predictions that are farther than $\delta$ from the ground truth (or the ground truth is occluded), and "false negatives" ($FN$) are ground truth points where the prediction is farther than $\delta$ (or occluded). Jaccard$_\delta$ is $TP/(TP + FP + FN)$, and Average Jaccard averages Jaccard$_\delta$ over several pixel thresholds. We use the same thresholds proposed in TAP-Vid (Doersch et al., 2022): namely, we resize all trajectories as if they had come from a $256 \times 256$ video, and use thresholds of 1, 2, 4, 8, and 16 pixels. We find that this model is quite accurate for real-world training data, with average points within threshold of 85.3 when evaluated on held-out data from the same distribution, and an average jaccard of 55.8, roughly meeting the performance of the underlying BootsTAPIR tracker.

**Architecture details**    For our transformer implementation, we use the standard design from Vaswani et al. (2017) with the pre-layer norm configuration from Xiong et al. (2020), and some of the additional improvements introduced in Dehghani et al. (2023). Specifically, we use the RMS norm applied to the keys and queries before computing attention weights, and execute self- and cross-attention paths (not the MLP path) in parallel. Another Layer Normalization layer (Ba et al., 2016) is applied to the output. Transformer hyperparameters are given in Table 9 with names given by their corresponding descriptions in the main text.

Additional hyperparameters for the sinusoidal positional embeddings, and the dimensionalities of projection operators, are given in Table 8.

| Sinusoidal embedding (number of frequencies) | 32 |
|---|---|
| Track token projection dimensionality ($C$) | 256 |
| Compression dimensionality | 64 |
| Up-projection dimensionality | 1024 - 128 |
| Query point encoder dimensionality | 1024 |

*Table 8.* Positional encoding and projection operator hyperparameters for TRAJAN.

| Transformer name | Attention type | QKV size | Layers | Heads | MLP size |
|---|---|---|---|---|---|
| Input track transformer | SA | $64\times8$ | 2 | 8 | 1024 |
| Perceiver-style tracks to latents | CA | $64\times8$ | 3 | 8 | 2048 |
| Up-projection latent transformer in decoder | CA | $64\times8$ | 3 | 8 | 2048 |
| Track readout transformer | CA | $64\times8$ | 4 | 8 | 1024 |

*Table 9.* Transformer architecture hyperparameters for TRAJAN. SA = self-attention, CA = cross-attention.

**Track motion radii calculation**    From the point tracks output by BootsTAPIR (Doersch et al., 2024), we can calculate two metrics of their overall motion: (1) their track lengths ($\sum_t \sqrt{(x_{t+1} - x_t)^2 + (y_{t+1} - y_t)^2}$), masking out non-visible tracks, and (2) their track radii (Figure 15), calculated as the radius of the smallest enclosing circle from the start to the end of a visible track. The track lengths will cover jittery movements or objects changing direction, while the motion radii will capture the maximum distance, in a consistent direction, for each point.

### B.2.2. MOTION HISTOGRAMS

Our implementation of motion histograms (Liu et al., 2024a) is based on `https://github.com/DSL-Lab/FVMD-frechet-video-motion-distance`. They propose to evaluate motion consistency in generated videos by estimating velocity and acceleration from point tracks. Inspired by HOG features (Dalal & Triggs, 2005), they partition the resulting volumes into $4 \times 5 \times 5$-sized tubelets and accumulate the magnitude of the values at each angle (using 8 bins) within a tubelet. Motion features are obtained by concatenating the resulting 1D histograms obtained for each tubelet using both velocity and acceleration. We apply this approach to 16-frame chunks of $64 \times 64$ densely sampled point tracks obtained from BootsTAPIR (Doersch et al., 2024) to yield a 9216-dimensional vector describing the motion within the corresponding 16-frame video.

### B.2.3. RAFT

To estimate optical flow we use the improved version (Sun et al., 2022) of RAFT (Teed & Deng, 2020) used as a baseline in Saxena et al. (2024). Since we are only concerned with obtaining the best possible flow model, we use a RAFT model which was trained on a large mixture of standard optical flow datasets. These include Sintel (Butler et al., 2012), KITTI (Geiger

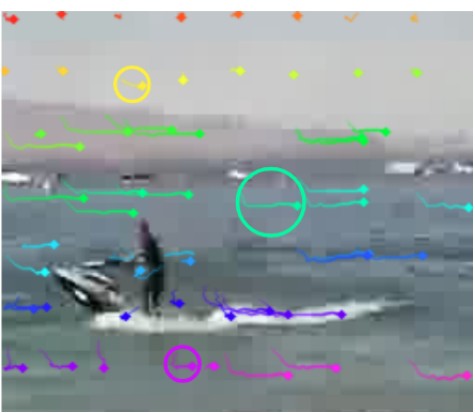

*Figure 15.* Examples of point trajectory radii. Circles are shown around maximum amount of motion. Radii are calculated from these circles.

et al., 2013), Kubric (Greff et al., 2022), TartanAir (Wang et al., 2020), FlyingThings (Mayer et al., 2016), and AutoFlow (Sun et al., 2021).

### B.2.4. MOOG

MooG (van Steenkiste et al., 2024) is a recurrent model trained for next-frame prediction. It operates by maintaining and updating an internal state constituting of off-the-grid latent tokens, which can be decoded to predict the next frame. These latents are first randomly intialized, and then updated on each iteration by a transformer model which cross attends to the image features of the corresponding frame, followed by a set of self-attention layers. The decoder converts the latent state back to pixels by querying the latents through cross-attention with fixed grid-based features (Sajjadi et al., 2022; Jaegle et al., 2022).

The original MooG implementation distinguishes between a "predicted" state and a "corrected" state, where the former only depends on the previous state, while the latter additionally integrates the current observation. To avoid the model only reconstructing the current observation, the loss is placed on the predicted state in this case. Here we slightly simplify the implementation by using placing the loss on the corrected state, but using the *next* observation when computing the loss. We train MooG for 600K steps on a mixture of datasets, including Kinetics-700 (Carreira et al., 2018), SSv2 (Goyal et al., 2017), ScanNet (Dai et al., 2017), Ego4D (Grauman et al., 2022), and Walking Tours (Venkataramanan et al., 2024).

### B.3. Human Study

To evaluate whether the proposed metrics are useful, we compare them to human judgements. Prior works mainly source human labels at a coarse-grained level using pair-wise comparisons between videos. For example, Unterthiner et al. (2018); Luo et al. (2025) ask humans to compare videos from two different sources (either different models, or with different levels of noise applied to them) to determine which of the two looked better, or alternatively report that their quality was indistinguishable. Liu et al. (2024a) take a similar approach using three pairwise comparisons: if a human expresses the same preference for at least two of the pairs then they are determined to prefer that model.

An alternative is to ask humans to evaluate individual videos as in Kim et al. (2024). There they use a 6-point scale to evaluate realism of videos, and a 3-point scale for motion or "temporal naturalness". Similarly, Bansal et al. (2024b) asks humans to indicate with yes/no whether videos "follow Physics Laws or Physical Commonsense". Liu et al. (2024c) asks humans to score individual videos based on their "motion quality" and "temporal consistency" using a 5-point scale.

A possible concern when sourcing human labels in this way is that the questions might be too open-ended, which limits their usefulness for evaluating motion in generated videos and for developing corresponding metrics. More generally, it isn't clear what motion-related dimensions are measured when asking about "physical commonsense" or when having humans express a preference for one generated video over another. To improve upon this, we propose to source fine-grained human annotations for generated videos using the questions detailed in Figure 16. To avoid conflating perfect consistency in videos that have motion with videos that contain no motion, we first ask humans about whether a video contains any motion at all, and only follow-up with the next two consistency related questions if they answer "Yes". Similarly, we first ask humans about whether interactions take place in a video. If they answer "Yes", then the slider for rating how realistic these interactions are appears. The other questions are available at all times and not conditioned on any of the previous answers.

We made use of a rater pool of 10 raters, and used 3 raters per question (randomly assigned). Before conducting the full human study, we asked raters to rate 10 questions, and gave feedback on their responses. For example, after an initial pilot, we noticed that many of the ratings were either 0 or 100 (both extreme endpoints of the slider), with few responses in between, and encouraged raters to make use of the full range of the slider to express degrees of agreement. We also noticed that camera motion was initially incorrectly evaluated, which we clarified. Feedback was also provided once we had obtained the full results for VideoPhy, before starting EvalCrafter. In particular, we noted that many videos with inconsistent motion (such as the objects or camera jittering / jumping around) received high scores for motion consistency. This is not desirable as for this data it can be assumed that the objects / camera are expected to move in a smooth and continuous manner, which we clarified. We encountered several videos with plausible looking individual frames, but that are stitched together in unnatural ways (no continuity between the frames in terms of content, see Figure 12a), which received high realism scores. In this case we provided feedback that this question concerns the entire video as a whole and not the sum of individual frames.

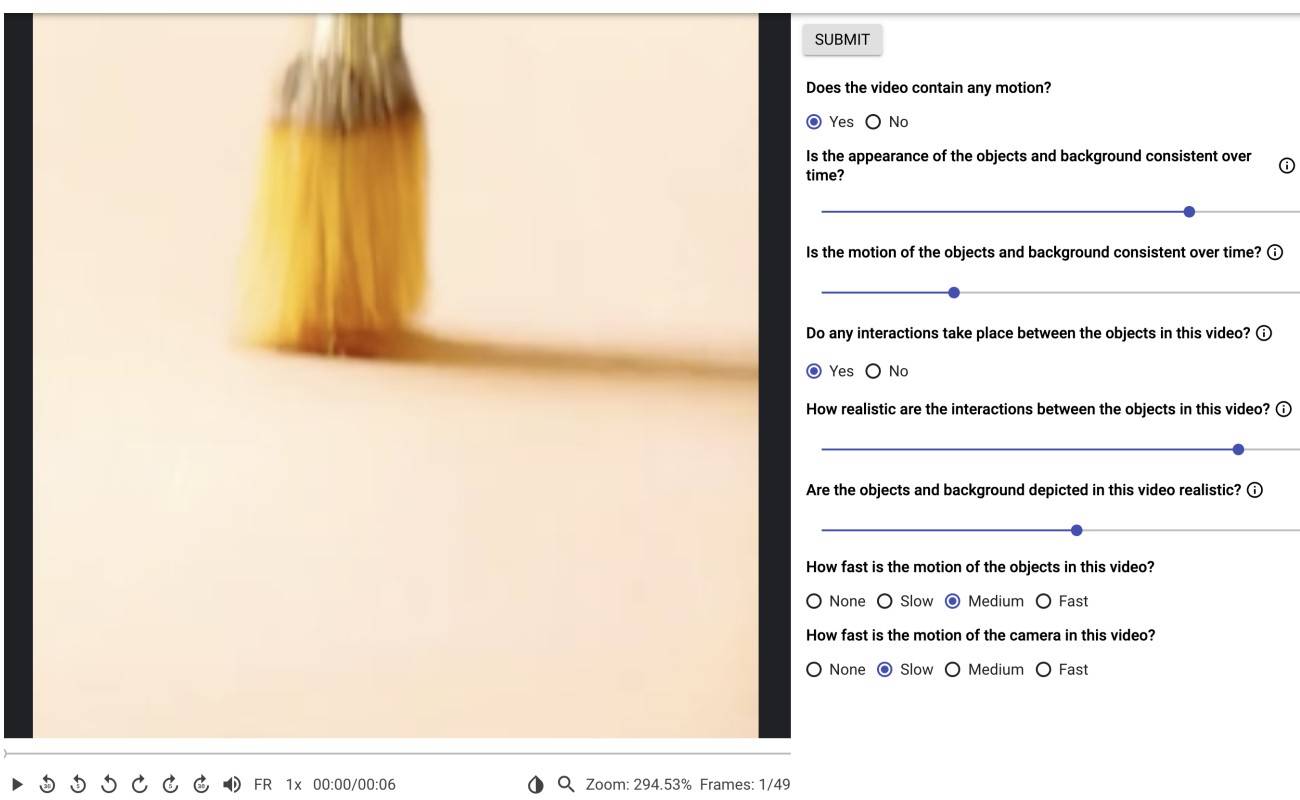

*Figure 16.* **A screenshot of the rater UI**. The video plays automatically on repeat as soon as the UI is opened. Videos were rendered at 8fps for ease of rating.

