# OpenReview forum: "Direct Motion Models for Assessing Generated Videos"
_ICML.cc/2025/Conference — ICML 2025 poster_

### Official Review · Reviewer_ANes · 2025-03-10

**Overall Recommendation:** 2

**Summary:**

This paper proposes TRAJAN, an architecture designed to obtain dense high-level motion features using tracks predicted by the BootsTAPIR model. The authors demonstrate that these extracted features can effectively measure pairwise distances between videos in terms of motion, as well as evaluate the temporal distortions within individual videos. While the authors conduct extensive experiments to showcase the effectiveness of the learned TRAJAN features, several claims remain inadequately supported. Additionally, the paper's presentation requires substantial improvement.

**Claims And Evidence:**

The claims in Section 5.2 are not all well-supported:
- The authors claim that "TRAJAN latent space is capturing something fundamentally different from pixel error" (line 352). This claim is supported by the evidence presented in Table 2.
- However, there appears to be a contradiction between interpretations of Figure 5. In Figure 5 (left), the authors claim "since the camera motion is correct, the distance between their TRAJAN embeddings is small." Yet in Figure 5 (right), even when camera motion is identical in two videos (both static), the TRAJAN difference is large. This raises an important question: is the TRAJAN embedding more sensitive to camera motion or subject motion?

Furthermore, Table 3 shows that the RAFT score outperforms TRAJAN-Len. in measuring object speed and performs comparably in evaluating camera speed. This finding potentially diminishes the value of the proposed metric.

**Essential References Not Discussed:**

N/A

**Experimental Designs Or Analyses:**

Please refer to the sections above.

**Methods And Evaluation Criteria:**

The authors conducted comprehensive experiments demonstrating correlations between the TRAJAN metric and human evaluation. However, they did not provide results on using the TRAJAN metric to compare existing video generation models. Such analysis would provide valuable insights into the performance of current methods and help researchers better understand the proposed metric's effectiveness.

**Other Comments Or Suggestions:**

The paper is poorly written. Below is a non-exhaustive list of issues identified during review:

- Figure Quality: Figure 1 is blurry and difficult to interpret, even after zooming in and carefully reading the captions. I suggest embedding videos into the PDF or including them in supplementary material for better clarity.
- Citation Inconsistency: The citation format varies throughout the paper. For example, one venue is listed as "ICLR," while another appears as "ICLR 2024-Twelfth International Conference on Learning Representations." A consistent citation format should be maintained.
Inconsistent Capitalization: Some subsection titles have the first letter capitalized (e.g., Sec. 3.2), while others do not (e.g., Sec. 5.1-5.3). Consistency in formatting is necessary.
- References: The SSv2 dataset is mentioned in the main paper but is only cited in the supplementary material. All references should be properly included in the main text.
- Formatting Issues: The spacing between Section 3.2 and Section 3.2.1 titles is irregular.
- Metric Descriptions: For TRAJAN-Len. and TRAJAN-Radii metrics, clear descriptions or direct links to their corresponding explanations should be provided rather than placing this information randomly in the Appendix.

**Other Strengths And Weaknesses:**

Please see comments above and below.

**Questions For Authors:**

The authors should provide more insights on their motivation. Is the proposed TRAJAN feature intended to serve as a standard metric to evaluate motion magnitude or plausibility for video generation models? The paper would benefit from clearer articulation of the broader purpose and application of this work.

This paper potentially offers insights to the community. However, based on the poor presentation and unsupported claims noted above, I recommend a weak reject.

**Relation To Broader Scientific Literature:**

The paper builds upon prior work in evaluating video generation models, focusing on measuring temporal distortions in generated videos.

**Theoretical Claims:**

N/A

---

> ### Author Rebuttal · Authors · 2025-04-01
>
> Thank you for your constructive comments and feedback.
>
> > there appears to be a contradiction between interpretations of Figure 5.
>
> TRAJAN is sensitive to differences in motion between videos (whether they be from a camera, or from an object). In Figure 5(a), the generated and reference video motions are the same for both the camera and the objects – the camera pans to the left in both cases, and none of the objects move independently of the camera, even though they have different appearances between the two videos. In Figure 5(b), the camera does not move, but the man moves in a distinctly different way. As a result, TRAJAN predicts a small distance in the first case (all motions are the same), and a large distance in the second case (the motion of the object is different). *There is therefore no contradiction in Figure 5.*
>
> > is the TRAJAN embedding more sensitive to camera motion or subject motion?
>
> We separately analyze how well TRAJAN captures human evaluations of generated videos when they have either high camera motion (but low subject motion) or high subject motion (but low camera motion) for the EvalCrafter dataset. The results are provided [here](https://sites.google.com/view/trajan-videos-anonymous/home) in the last section.
>
> __In both settings, TRAJAN outperforms all baselines__. TRAJAN also captures human judgments better in the high camera motion vs. high object motion setting (motion consistency 0.53 vs. 0.30; appearance consistency 0.56 vs. 0.29; realism 0.45 vs. 0.24). However, we also found that human raters were less consistent within the high object motion setting, suggesting that this may be a generically harder task (for both humans and models).
>
> > Table 3 shows that the RAFT score outperforms TRAJAN-Len. in measuring object speed (...). This finding potentially diminishes the value of the proposed metric.
>
> > The authors should provide more insights on their motivation. Is the proposed TRAJAN feature intended to serve as a standard metric to evaluate motion magnitude or plausibility for video generation models? The paper would benefit from clearer articulation of the broader purpose and application of this work.
>
> The proposed metric was not primarily intended for measuring object and camera speed. We included this result as an interesting addition to the __main arguments of the paper__: __TRAJAN is useful as a metric for measuring motion quality and motion differences in videos by simultaneously providing a compact latent space representation of motion and reconstruction errors for evaluating individual videos__.
>
> Importantly, in this work, and for the first time, we study metrics across distribution-level comparisons, video-video comparisons and evaluations of individual videos in terms of their motions. This involved adapting several prior methods, such as I3D, to the per-video setting as well as exploring new approaches, such as MooG, for evaluating videos. We also conducted a human study that includes fine-grained motion categories for evaluating metrics. All in all we identify numerous shortcomings of existing approaches (including RAFT) and propose a new metric based on point tracks (TRAJAN) that works well across all of the settings we tested.
>
> __To our knowledge, TRAJAN is the first metric that can operate across the distribution-, video-video comparison-, and single-video evaluation- levels, providing a compelling metric for motion in all cases.__
>
> > The authors conducted comprehensive experiments [...]. However, they did not provide results on using the TRAJAN metric to compare existing video generation models.
>
> Our human evaluations on VideoPhy and EvalCrafter are using video samples from existing video generation models. Based on your feedback we have recomputed the results from Table 1 comparing human judgements and metric scores separately for each model and then aggregating. __We found that TRAJAN correlates significantly better with human evaluations of which video model creates the most realistic videos: 0.76 (TRAJAN) 0.53 (RAFT) 0.43 (VideoMAE) 0.51 (I3D) 0.58 (MooG)__.
>
> > The paper is poorly written. Figure Quality. Citation Inconsistency. Capitalization. References. Formatting Issues. Metric Description.
>
> We were surprised that you found this to be the case and listed “poor presentation” as one of the reasons for recommending weak reject. The examples you listed, such as citation forms, subsection capitalization, and spacing, can be easily addressed for a camera-ready paper. We will ensure the camera-ready version of our manuscript will have each of these addressed. Thank you for pointing these out.
>
> We provide videos for Figure 1, with additional experiments, [here](https://sites.google.com/view/trajan-videos-anonymous/home). We hope that by ensuring that the formatting concerns are addressed in a camera-ready version of the manuscript, the “comprehensive evaluations” and “insights to the community” that you highlight might lead you to reconsider your score.

---

> > ### Comment · Reviewer_ANes · 2025-04-03
> >
> > I appreciate the authors' rebuttal, which addresses some of my concerns. However, a significant issue remains unaddressed: the absence of TRAJAN metric results comparing existing video generation models. For a metric intended to evaluate video generation models, it is essential to demonstrate how existing models (such as SVD, CogVideoX, Hunyuan, and Cosmos) perform on this benchmark. Without such comparisons, it's difficult to assess whether the proposed metric aligns with users' experience of these models.
> >
> > Furthermore, the metric's inability to differentiate between camera motion and subject motion raises questions about its utility to the video generation community. This limitation significantly impacts its interpretability. Consider, for example, the well-known "Tokyo Walk" video generated by Sora, which incorporates both camera and subject motion.
> >
> > I am also **surprised** by the **numerous errors** throughout the manuscript. Such mistakes suggest insufficient proofreading and a lack of attention to detail. In my humble opinion, rigorous scientific work demands a clear presentation. The issues I identified were easily noticeable even upon a cursory review, indicating they could have been readily corrected with careful proofreading.

---

> > > ### Author Response · Authors · 2025-04-04
> > >
> > > > “However, a significant issue remains unaddressed: the absence of TRAJAN metric results comparing existing video generation models. For a metric intended to evaluate video generation models, it is essential to demonstrate how existing models (such as SVD, CogVideoX, Hunyuan, and Cosmos) perform on this benchmark. Without such comparisons, it's difficult to assess whether the proposed metric aligns with users' experience of these models.”
> > >
> > > We draw your attention to section 5.3 with results on the following video generation models:
> > >
> > > __within EvalCrafter__: MoonValley, ZeroScope, Floor33, Gen2, HotShot, ModelScope, Pika (versions 1 and 2), Show-1, VideoCrafter-1.
> > >
> > > __within VideoPhy__: __CogVideo-X-2B__, __CogVideo-X-5B__, LumaAI, Gen2, LaVie, OpenSora, Pika, __SVD__, VC2, ZeroScope.
> > >
> > > This covers 17 different video generation models. We show in all of our results  that the TRAJAN metric captures a significant fraction of the human variance in how these models are rated for individual videos. This is exactly the users’ experience of these models – we directly ask users to rate videos generated by each of these video models, and show that our metric correlates well with their responses. For our rebuttal, we also showed that the metric captures the users’ rankings in which video generation model performs best. Note that this exactly tracks norms in the field: see e.g. EvalCrafter Figure 4 and Table 2.
> > >
> > > > Furthermore, the metric's inability to differentiate between camera motion and subject motion raises questions about its utility to the video generation community. This limitation significantly impacts its interpretability. Consider, for example, the well-known "Tokyo Walk" video generated by Sora, which incorporates both camera and subject motion.
> > >
> > > We show that TRAJAN can capture human ratings of realism and consistency for __both subject motion and camera motion__, both in isolation and together, going beyond prior work like SIFT-Sora which can only be applied to videos with camera motion. The goal of the metric is not to differentiate what kind of motion is occurring (a classification problem), but rather whether the motion is realistic or not. The metric would directly apply to the Tokyo Walk video, and likely rate it very highly, because the motion of both the subject and the camera is smooth and realistic.
> > >
> > > > I am also surprised by the numerous errors throughout the manuscript. Such mistakes suggest insufficient proofreading and a lack of attention to detail.
> > >
> > > The issues that you point to are primarily minor formatting mistakes and “numerous errors” is an exaggeration. For example, there is *only one subsection* with incorrect capitalization. All others are capitalized because they are proper names (of methods, or metrics). Similarly, *only one subsection* has incorrect spacing and there is *only one instance* of a dataset that was only cited in the Appendix but not in the main text. Other remarks such as that information is placed “[...] randomly in the Appendix” is subjective, especially considering its subtitle in the Appendix is “Track motion radii calculation”, an effort to make sure it was easily found by readers. Issues with citation consistency are also common for conference papers in machine learning, for example the references section of this [*this best paper award winner  in ICML last year*](https://openreview.net/pdf?id=bJbSbJskOS). We are happy to correct these issues for the camera-ready paper, but again, we point to the fact that none of the other reviewers expressed any concern with the presentation or quality of the results.
> > >
> > > >This calls into question the authors' commitment to their own work. When authors appear not to respect their own research, it becomes difficult for reviewers and readers to accord it the respect it might otherwise deserve.
> > >
> > > This is an unprofessional and unproductive remark. We put a significant amount of time into the presentation of this work, including the rebuttal, as evidenced by the clear interpretation of the work by the other two reviewers.

---

### Official Review · Reviewer_P3xr · 2025-03-14

**Overall Recommendation:** 3

**Summary:**

This paper proposes TRAJAN, a novel motion-focused evaluation framework for assessing the quality of generated videos. Unlike traditional metrics like FVD that emphasize appearance, TRAJAN uses auto-encoded point tracks to assess temporally extended motion features. It supports distribution-level, video-pair, and per-video evaluation. Experiments show TRAJAN achieves higher sensitivity to temporal distortions and better alignment with human judgments across benchmarks (EvalCrafter, VideoPhy), outperforming existing motion and appearance-based metrics.

**Claims And Evidence:**

The paper makes several key claims:

1. TRAJAN provides better sensitivity to temporal distortions than prior metrics like FVD, VideoMAE, or I3D.
2. TRAJAN embeddings correlate better with human ratings of motion consistency, realism, and interaction quality.

All claims are thoroughly validated by extensive empirical evaluation.

**Essential References Not Discussed:**

The paper comprehensively reviews prior work. No critical omissions were identified.

**Experimental Designs Or Analyses:**

The experiments are rigorous and cover multiple evaluation modalities:

1. Controlled distortions on UCF101 to measure temporal sensitivity.
2. Video-to-video comparisons using WALT.
3. Per-video quality assessments using human ratings from EvalCrafter and VideoPhy.

**Methods And Evaluation Criteria:**

The TRAJAN model architecture is methodologically sound and well-motivated. It uses a Perceiver-style transformer to encode dense point tracks and reconstruct them from query points.
Comparisons with other motion-based (e.g., RAFT warp error, motion histograms) and appearance-based (e.g., VideoMAE, I3D) metrics are comprehensive and insightful.

**Other Comments Or Suggestions:**

N.A.

**Other Strengths And Weaknesses:**

Weaknesses:
1. TRAJAN performance still misses edge cases where semantic context is crucial (e.g., object disappears unrealistically).
2. Reliance on point tracks may limit applicability in occlusion-heavy or textureless scenes.

**Questions For Authors:**

N.A.

**Relation To Broader Scientific Literature:**

This work fills a critical gap in the video generation literature by emphasizing motion quality assessment over frame-based metrics.

**Theoretical Claims:**

The paper does not introduce formal theoretical analysis but provides strong conceptual justification for modeling motion independently from semantics via point track encoding.

---

> ### Author Rebuttal · Authors · 2025-04-01
>
> Thank you for your constructive comments and feedback. We were pleased to see that you found our approach “methodologically sound and well-motivated” and that “All claims are thoroughly validated by extensive empirical evaluation”.
>
> You also mentioned how “The paper [...] provides strong conceptual justification for modeling motion independently from semantics via point track encoding.” and how  “This work fills a critical gap in the video generation literature by emphasizing motion quality assessment over frame-based metrics.”
>
> >“TRAJAN performance still misses edge cases where semantic context is crucial”, “Reliance on point tracks may limit applicability in occlusion-heavy … scenes”
>
> We highlight and discuss this first point about semantic context in Figure 6, where __none__ of the metrics can capture the unexpected disappearance of the beer glass. Fully solving this challenge would require learning a fully accurate world model. This is an exciting future direction, but it is outside the scope of our work.
>
> Occlusion-heavy scenes are similarly challenging for all existing automated metrics which we compare to here (optical flow, action recognition, and masked auto-encoding will similarly suffer in occlusion-heavy scenes, although possibly less so than point tracking).
>
> >“Reliance on point tracks may limit applicability in … textureless scenes”
>
> Working with textureless scenes is a general challenge for point tracking. In practice, many of the videos we evaluate are textureless. In such cases, the base point tracker does produce relatively poor point tracks which can drift and be inappropriately marked as occluded. However, because TRAJAN is trained to do point track *reconstruction*, it can *learn* this from data, and the reconstruction errors on textureless scenes are still, on average, reasonably small. __Some examples of this can be seen  [here](https://sites.google.com/view/trajan-videos-anonymous/home) (second section) where point tracks for textureless parts of the background are still well reconstructed (shown in blue)__. This is an advantage of using a trained autoencoder over the point tracks instead of calculating a metric directly on the point tracks themselves. Thank you for raising this point, we will add a discussion of this to the paper.

---

### Official Review · Reviewer_zaWZ · 2025-03-14

**Overall Recommendation:** 3

**Summary:**

The authors propose a new video evaluation metric using point tracks instead of pixel reconstruction or recognition features, which can evaluate temporal consistency.

**Claims And Evidence:**

The claims seem to be supported well in this paper.

**Essential References Not Discussed:**

N/A/

**Experimental Designs Or Analyses:**

The experimental design is valid but the corrupted images as demonstrated in figure.4 do not seem to be low-level distorted (like blurred or locally rotated and shifted). That casts doubt on whether this proposed TRAJAN metric can indeed capture the distortions of real motions.
Some other methods also include motion distortion such as EvalCrafter [1] , VBench [2], or SIFT-sora [3]. This paper should compare against those methods.

[1] Liu, Yaofang, et al. "Evalcrafter: Benchmarking and evaluating large video generation models." Proceedings of the IEEE/CVF Conference on Computer Vision and Pattern Recognition. 2024.

[2] Huang, Ziqi, et al. "Vbench: Comprehensive benchmark suite for video generative models." Proceedings of the IEEE/CVF Conference on Computer Vision and Pattern Recognition. 2024.

[3] Sora Generates Videos with Stunning Geometrical Consistency

**Methods And Evaluation Criteria:**

The evaluation makes sense that demonstrates that this proposed metric is better at capturing motion distortions.

**Other Comments Or Suggestions:**

N/A/

**Other Strengths And Weaknesses:**

See above.
Some work like [3] works on pure geometry for estimating motion score, but this work still uses representation learning, which lacks explainability a little. Including more analysis on score interpretation will be helpful.

**Questions For Authors:**

See above.

**Relation To Broader Scientific Literature:**

This work can be impactful if they have a good motion quality assessment method as validated thoroughly.

**Theoretical Claims:**

N/A

---

> ### Author Rebuttal · Authors · 2025-03-31
>
> Thank you for your constructive comments and feedback. Please see [https://sites.google.com/view/trajan-videos-anonymous/home](https://sites.google.com/corp/view/trajan-videos-anonymous/home) for additional experiments on score interpretation.
>
> > “The corrupted images as demonstrated in figure.4 do not seem to be low-level distorted (like blurred or locally rotated and shifted). That casts doubt on whether this proposed TRAJAN metric can indeed capture the distortions of real motions”
>
> The distortions in Figure 4 are taken from prior work by [Ge et al. (2024)](https://arxiv.org/abs/2404.12391) which were chosen to highlight challenges faced by existing metrics (like FVD) in addressing *temporal* above and beyond *appearance-based* distortions. These distortions include local noise (Corruption 2.3, Figure 4e) and local warping (Corruption 1.2, Figure 4b) and were designed to capture realistic distortions from cameras or other sensors. __This experiment primarily highlights the advantage of TRAJAN over prior work (including Ge et al, 2024) for being sufficiently sensitive to temporal rather than appearance-based distortions (i.e. motion biased), and we therefore matched the experimental conditions from prior work__.
>
> However, we agree with the reviewer that synthetic distortions are not necessarily representative of real motion distortions. This informed our choice to investigate different model’s sensitivities to the kinds of distortions introduced by state of the art video generation methods.
>
> Our evaluations on VideoPhy and EvalCrafter (as referenced in your review) are doing precisely this. For example, in Appendix B.1.3 we describe how “[...] we randomly select 104 generated videos from 11 of the text-to-video models (the original 5 from the EvalCrafter human evaluation dataset, and 6 additional models).” to conduct a human study. __In Table 1 and 2, we report how TRAJAN is the top-performing model that best correlates with human judgements of motion/appearance consistency and realism.__
>
> >”Some other methods also include motion distortion such as EvalCrafter [1] , VBench [2], or SIFT-sora [3]. This paper should compare against those methods.”
>
> The __RAFT-Warp error__, which we report in our paper, __is the one used in EvalCrafter, with a related RAFT-based metric in VBench for assessing motion quality and distortions (“Temporal Quality - Dynamic Degree”)__. Across all of our experiments, __TRAJAN performs better than or equal to RAFT-Warp for assessing motion quality and realism__ including on the EvalCrafter dataset.
>
> SIFT-sora is a compelling method for interpretability, but can *only* be applied to videos where the motion solely comes from the camera. __If any of the objects move independently of the camera (which is the case in many generated videos), SIFT-sora cannot be applied__.
>
> > “Some work like [3] works on pure geometry for estimating motion score, but this work still uses representation learning, which lacks explainability a little. Including more analysis on score interpretation will be helpful.”
>
> Thank you for this suggestion – we have added a detailed investigation of whether TRAJAN scores can provide more interpretability. Since TRAJAN measures the reconstruction of individual point tracks (Figure 1), it is possible to localize errors in both space and time (with badly reconstructed tracks, shown in red, indicating where the largest errors are). __We provide 3 further examples of this kind of score interpretation [here](https://sites.google.com/view/trajan-videos-anonymous/home)__.
>
> In each example, we show on the left the full video, in the center the Average Jaccard across all points for each frame of the video, and on the right a clip from the video centered on the frame with the worst overall Average Jaccard. __In the first two cases, there is a major change in object appearance partway through the video, which is picked up in the time course plots. Looking at individual points, the greatest errors are centered on the part of the video that unnaturally changes in appearance, as indicated by the red and white colored point tracks.__ In the last case, this video maintains consistent motion throughout, as evidenced by the consistently high Average Jaccard. Most points are well reconstructed, with the only exceptions being points on the border of the moving object, which are generally ambiguous (their motions could go either way).
>
> We will update the paper to discuss this aspect of our approach in greater detail and include more examples of localizing motion errors in this way.

---

> > ### Comment · Reviewer_zaWZ · 2025-04-02
> >
> > The authors address my concern well, and I plan to improve my rating.

---

### Decision · Program_Chairs · 2025-05-01

**Decision:**

Accept (poster)

**Comment:**

This paper presents work on assessing video generation methods.  The paper had divergent reviews after the rebuttal phase.  On the positive side, the proposed TRAJAN model architecture is methodologically sound and well-motivated.  Comprehensive comparisons with other motion-based and appearance-based metrics are made, from a perspective of alignment to human judgements.

The main negatives raised by all the reviewers included details over the method, with its potential for sensitivity to camera motion or subject motion.  Importantly, there was a concern over lack of comparison of existing video generation models: does the proposed metric provide insights on video generation.

On balance, the paper does produce a state of the art method for evaluating video generation, which is demonstrated via alignment to human evaluations.  While there are concerns over the generality and the aspects of the videos the method is measuring, it does represent a step forward in machine learning for video generation.